# Genomic mutation landscape of skin cancers from DNA repair-deficient xeroderma pigmentosum patients

Andrey A. Yurchenko [1,10], Fatemeh Rajabi[1,10], Tirzah Braz-Petta[2], Hiva Fassihi [3], Alan Lehmann[3,4], Chikako Nishigori[5], Jinxin Wang[1], Ismael Padioleau [1], Konstantin Gunbin[1], Leonardo Panunzi[1], Fanny Morice-Picard[6], Pierre Laplante[1], Caroline Robert [1,7], Patricia L. Kannouche [8], Carlos F. M. Menck [9], Alain Sarasin[8] & Sergey I. Nikolaev [1] ✉

Xeroderma pigmentosum (XP) is a genetic disorder caused by mutations in genes of the Nucleotide Excision Repair (NER) pathway (groups A-G) or in Translesion Synthesis DNA polymerase η (V). XP is associated with an increased skin cancer risk, reaching, for some groups, several thousand-fold compared to the general population. Here, we analyze 38 skin cancer genomes from five XP groups. We find that the activity of NER determines heterogeneity of the mutation rates across skin cancer genomes and that transcription-coupled NER extends beyond the gene boundaries reducing the intergenic mutation rate. Mutational profile in XP-V tumors and experiments with *POLH* knockout cell line reveal the role of polymerase η in the error-free bypass of (i) rare TpG and TpA DNA lesions, (ii) 3' nucleotides in pyrimidine dimers, and (iii) TpT photodimers. Our study unravels the genetic basis of skin cancer risk in XP and provides insights into the mechanisms reducing UV-induced mutagenesis in the general population.

Xeroderma Pigmentosum (XP) is a group of eight rare hereditary recessive disorders caused by mutations in seven nucleotide excision repair (NER) pathway genes (groups A-G) or in the *POLH* gene coding the translesion synthesis (TLS) DNA polymerase η (XP-V)[1]. XP is characterized by up to a 10000-fold increased risk of non-melanoma skin cancers and 2000-fold increased risk of melanoma[2]. Moreover, epidemiological studies revealed a 34-fold increased risk of internal tumors in XP patients which was associated with characteristic mutation signature and accelerated accumulation of mutations[3,4].

Nucleotide excision repair (NER) is the main pathway that removes bulky DNA lesions in the genome in an error-free manner[5]. NER can be initiated by two sub-pathways: global genome repair (GG-NER) and transcription-coupled repair (TC-NER), while the downstream mechanism of lesion removal is shared between the two and involves recruitment of the TFIIH complex and XPA, which unwind the DNA helix at the lesion site, and XPG and XPF-ERCC1, which excise the fragment containing the damaged nucleotides[6]. GG-NER operates genome-wide; it recognizes UV-induced bulky lesions with the

[1]INSERM U981, Gustave Roussy Cancer Campus, Université Paris Saclay, Villejuif, France. [2]Departamento de Biologia Celular e Genética, Universidade Federal do Rio Grande do Norte, Av. Senador Salgado Filho, s/n, Natal 59078-970, Brazil. [3]National Xeroderma Pigmentosum Service, Department of Photodermatology, St John's Institute of Dermatology, Guy's and St Thomas' Foundation Trust, London SE1 7EH, UK. [4]Genome Damage and Stability Centre, University of Sussex, Falmer, Brighton BN1 9RQ, UK. [5]Division of Dermatology, Department of Internal Related, Kobe University Graduate School of Medicine, Kobe, Japan. [6]Service de Dermatologie, CHU de Bordeaux, Bordeaux, France. [7]Department of Medical Oncology, Gustave Roussy and Paris-Saclay University, Villejuif, France. [8]CNRS UMR9019 Genome Integrity and Cancers, Institut Gustave Roussy, Université Paris-Saclay, Villejuif, France. [9]Department of Microbiology, Institute of Biomedical Sciences, University of Sao Paulo, Sao Paulo, SP, Brazil. [10]These authors contributed equally: Andrey A. Yurchenko, Fatemeh Rajabi. ✉e-mail: sergey.nikolaev@gustaveroussy.fr

XPE/DBB2 protein or helix distortions caused by those lesions with XPC protein. TC-NER is initiated by lesion-stalled RNA polymerase II and operates mainly on the transcribed strand of active genes.

The photoproducts which have not been removed by NER may block the progression of replicative polymerases during DNA replication. The TLS polymerase η is a DNA polymerase that bypasses the UV-induced photoproducts, thus preventing replication fork stalling[7,8].

Skin cancer predisposition among XP groups is highly heterogenous, and there is an inverse relationship between the level of sunburn sensitivity and skin cancer incidence between the groups[9]. The most skin cancer-prone groups are XP-C and XP-E, with impaired GG-NER, and XP-V—with the deficiency of polymerase η[9]. Other XP groups with the deficiency in both GG-NER and TC-NER are also associated with considerable skin cancer risk and demonstrate a high level of sunburn sensitivity and neurological symptoms[10]. A current model posits that UV exposure in the context of TC-NER deficiency may cause an impairment of transcription and result in decreased cell fitness, whereas defective GG-NER results in tumor proneness[11].

The process of UV-induced mutagenesis depends on several major factors, including DNA lesion generation, removal by NER, and bypass by TLS polymerases. Skin cancers from XP groups differ from each other and sporadic skin cancers by the ability to repair or bypass DNA lesions, but not by the sources of DNA damage. Thus, analysis of XP tumors with defects in GG-NER alone, both TC- and GG-NER, or TLS, enables the disentanglement of the contribution of those components to mutagenesis in a natural physiological system. Moreover, an extreme skin cancer susceptibility in XP patients points to vulnerabilities in the mechanisms of protection from excessive mutation accumulation in normal skin cells.

In this work, we assemble a collection of 38 skin cancers from 5 xeroderma pigmentosum groups (XP-A, C, D, E, V). We use Whole Genome Sequencing (WGS) to study the role of defects in the major components of NER and translesion synthesis on tumor mutation burden, mutation profiles, genomic landscape, and protein-damaging effects of mutagenesis in human skin cancers.

## Results

### Samples and clinical characteristics
We collected and sequenced genomes of 33 skin cancers from 21 patients representing 5 out of 8 xeroderma pigmentosum (XP) groups (3 XP-A, 4 XP-C, 2 XP-D, 10 XP-E, and 14 XP-V tumors; Supplementary Table 1). Causative homozygous ($n = 12$) or compound heterozygous ($n = 8$) germline variants were identified in 20 patients, 13 of which had known causative germline mutations (Supplementary Table 1), while the 7 others—had novel germline mutations compatible with the diagnosis. The mean tumor purity and sequence coverage were 41% and 40× (30× for normal tissue), respectively. In addition, we sequenced genomes of 6 sporadic cutaneous squamous cell carcinoma samples (SCC). This newly generated data was combined with WGS data from four previously published XP-C tumors[12,13], one XP-D[14], as well as 25 sporadic cutaneous Squamous Cell Carcinomas (SCC)[12,15,16], 8 Basal Cell Carcinomas[15] (BCC) and 113 Melanomas[17] (MEL) from individuals not affected with XP. The resulting cohort of XP tumors included 17 BCCs, 15 SCCs, five melanomas, and one angiosarcoma. The mean age at biopsy in XP-cohort was 33 years old (ranging from 25 years old in the XP-C group to 48 years old in the XP-V group) while in sporadic skin cancer group it was 65 years old (Table 1, Supplementary Table 1).

### XP groups demonstrate different mutation burden and mutation profiles
We assessed the Tumor Mutation Burden (TMB) and mutation profiles of skin cancer genomes from 5 sequenced XP groups and compared them with the three types of sporadic skin cancer including BCC, SCC and MEL (Fig. 1a). The mean TMB of single base substitutions (SBS) was

significantly higher in 3 XP groups: XP-E (350 mut/Mb, $p = 0.0241$), XP-V (248 mut/Mb, $P = 0.0014$) and XP-C skin cancers (162 mut/Mb, $P = 0.0220$), than the dataset weighted average (130 mut/Mb, global $P < 2.2e−16$; Kruskal−Wallis $H$ test; Fig. 1a). We also observed a strong difference in the TMB and the proportion of CC > TT double base substitutions (DBS) characteristic of UV-induced mutagenesis between the different XP groups and sporadic cancers (Fig. 1a). The highest proportion of CC > TT DBS from UV-induced SBS in pyrimidine dimers (C > T in YpC or CpY contexts; Y denotes a pyrimidine) was observed in XP-C and XP-D tumors (0.2 and 0.17, respectively), which was 6 times higher than in sporadic skin cancers (0.03, $p = 4.7e−08$, Mann−Whitney $U$ test, two-sided).

The mutation profiles of skin cancers in all XP groups were dominated by C > T substitutions at pyrimidine dimers, as also found in sporadic skin cancers. However, some XP groups demonstrated marked differences for C > T mutations in specific contexts, such as enrichment at TCA in XP-E, TCW in XP-C, or NCY in XP-D (where W denotes A or T; N: A, C, G, or T; Y: C or T). Moreover, in XP-V skin cancers, we report abundant mutations, namely C:G > A:T, T:A > A:T, and T:A > C:G, which were not previously seen to a significant degree in skin cancer (Fig. 1b, Supplementary Fig. 1). XP tumors formed clusters by XP group, which were non-overlapping with the cluster of sporadic skin cancers based on SBS mutation profiles and multidimensional scaling analysis (MDS; Fig. 1c–e; Supplementary Figs. 2, 3). XP-V, XP-C, and XP-A clusters were located distantly, while the XP-E / XP-D cluster was closer to the cluster of sporadic skin cancers.

Among 78 COSMIC mutation signatures[18] (v3.2) extracted from the pan-cancer dataset, four mutation signatures (SBS7a/b/c/d) are associated with UV irradiation, and combination of SBS7a and SBS7b usually explain the majority of mutations in sporadic melanomas[18]. We investigated whether these signatures could explain the observed mutation profiles in XP skin cancer with an accuracy comparable to sporadic skin cancers. For that, we compared observed and reconstructed mutation profiles for each sample in our cohort. The mean Cosine dissimilarity distance was small for sporadic skin cancers (0.004) but increased drastically for all the XP groups (0.16) and particularly for XP-C (0.237), XP-A (0.1957), and XP-V (0.222, Fig. 1f, Supplementary Fig. 4) indicating that mutational profiles in XP skin cancer cannot be optimally reconstructed with the known UV mutational signatures.

### Nucleotide excision repair efficiency determines mutation load distribution along the genome
Strong heterogeneity in the mutation rate across the genome is an important fundamental feature of mutagenesis, which has several clinical implications, for example, the discovery of cancer driver genes. We investigated the distribution of typical UV mutations (YC > YT or CY > TY) in XP and sporadic skin cancers in relation to replication timing (RT), active and inactive topologically associated domains (TAD), and markers of chromatin states. These analyses revealed a major role for NER in shaping the heterogeneity of local rates of UV-induced mutations across the genome. A maximal 5.2-fold difference was observed between the earliest and the latest replicating bins in sporadic skin cancers (average for BCC, cSCC, and MEL) with a monotonal decrease of mutation load from late to early replicating genomic regions (Fig. 2a). This effect was much weaker in GG-NER deficient XP-C genomes (2.4-fold) and almost disappeared in GG-NER and TC-NER deficient XP-A (1.5-fold) and XP-D (0.99-fold) genomes (Fig. 2a). Interestingly, the distribution of UV-induced SBS by RT in XP-E and XP-V genomes was not very different from sporadic skin cancer genomes, 4.6-fold and 5.4-fold, respectively.

It has been recently shown that TAD boundaries between active and inactive chromatin domains strongly delineate the transition between regions with low and high mutation load in different human cancers[19]. Indeed, in our cohort, we found a 2.2-fold difference in

**Table 1 | The studied dataset of XP and sporadic skin cancers with WGS data**

| Group | Tumors (n) | Patients (n) | Mean age at biopsy (years) | cSCC (n) | BCC (n) | Melanoma (n) |
|---|---|---|---|---|---|---|
| XP-E | 10 | 4 | 28 | 7 | 2 | 1 |
| XP-C | 8[a] | 8 | 25 | 5 | 1 | 1 |
| XP-A | 3 | 1 | 32 | 1 | 2 | – |
| XP-D | 3 | 3 | 30 | 2 | 1 | – |
| XP-V | 14 | 9 | 48 | – | 11 | 3 |
| Sporadic SCC | 31 | 31 | 73 | 31 | – | – |
| Sporadic BCC | 8 | 8 | 66 | – | 8 | – |
| Sporadic MEL | 113 | 113 | 57 | – | – | 113 |

[a]- one XP-C patient with angiosarcoma.

mutation load between active and inactive TADs in sporadic cancers, but it was noticeably decreased in XP-C (1.4-fold) cancers and was virtually absent in XP-A (1.05-fold) and XP-D (1.09-fold; Fig. 2b). Similarly, the mutation load in XP-A and XP-D tumors was independent of chromatin states, the XP-C group demonstrated a mild dependence, while the XP-E and the XP-V groups were not different from sporadic cancers (Fig. 2c).

CPD and 6-4PP DNA lesions occur on pyrimidine bases, which enabled us to identify the strand on which the lesion underlying a UV-induced mutation occurred. In order to separately investigate the genomic targets of GG-NER and TC-NER, we split the genome into intergenic, transcribed, and untranscribed strands of genic regions. A strong decrease of mutation rate in the early RT regions in groups proficient in GG-NER (sporadic cancers and XP-V), and surprisingly in GG-NER deficient XP-E, was observed in intergenic regions and untranscribed strands of genes. Whereas XP-A and XP-D which lack both GG-NER and TC-NER, had flat slopes compatible with the lack of repair in the open chromatin of early RT regions (Fig. 2d, Supplementary Fig. 5). XP-C samples with the functional TC-NER and fully abrogated GG-NER demonstrated lack of repair on untranscribed gene strands and in the intergenic regions, but they were proficient in repair of the transcribed strands of genes (Fig. 2d, Supplementary Fig. 5).

**Transcriptional bias is different between the XP groups**

TC-NER removes UV-induced bulky DNA lesions on the transcribed strand of expressed genes more efficiently than GG-NER on the untranscribed strand resulting in a decrease of mutations on the transcribed versus untranscribed strand, a phenomenon called transcriptional bias (TRB)[20]. In skin tumors with proficient NER, the TRB ranged between 1.3 and 1.6-fold for sporadic cancers and was 1.7 in XP-V. In the GG-NER-deficient TC-NER-proficient groups, TRB was particularly high, ranging between 1.77-fold (XP-E) and 2.42-fold (XP-C), which is compatible with defects in the repair of the untranscribed strand. In contrast, in XP-A and XP-D groups with defects of both TC-NER and GG-NER TRB was minimal or absent: 1.17-fold and 0.97-fold, respectively (Fig. 3a).

**TC-NER removes DNA lesions downstream of genes and influences intergenic mutation load**

Since early RT regions are particularly gene-rich[21], we hypothesized that in GG-NER deficient, but TC-NER proficient XP groups (XP-C, XP-E), decreased mutation load in early RT intergenic regions might be associated with the TC-NER activity beyond gene boundaries. Indeed, in GG-NER deficient XP-C tumors, we revealed a statistically significant TRB up to 40 kb downstream of the furthest annotated transcriptional end sites (TES) of genes with decreased mutation frequency on the transcribed strand of nearby genes (Fig. 3b, Supplementary Fig. 6). The same effect was observed in XP-E and even in NER proficient skin cancers although with a lower magnitude (but significant for sporadic

melanoma with the large sample size, Supplementary Fig. 6). As expected, we did not observe TRB downstream of genes in XP-A and XP-D samples being deficient for both TC-NER and GG-NER (Fig. 3b, Supplementary Fig. 6). To validate TC-NER activity downstream of gene TES, we used previously published XR-seq data from XPC-deficient cell lines[22,23]. It is expected that in XPC-deficient cells, XR-seq data, representing the sequencing of lesion-containing DNA fragments excised by NER[22], is produced exclusively by TC-NER. An XR-seq signal was observed up to 40 kb downstream of TES on a transcribed strand of a nearby gene, mirroring mutation asymmetry in the same regions in XP-C tumors (Fig. 3c, Supplementary Fig. 7) and was well correlated with the transcriptional intensity of nascent RNA, which was retrieved from an independent study[24] (Fig. 3d). This suggests that, in some cases, the RNA polymerase might continue transcription after TES and recruit TC-NER at lesion sites. We identified XR-seq signal in 21% of the cumulative length of intergenic regions and 14%−of untranscribed strands of genes in XPC-deficient cell line[22], suggesting ubiquitous extended TC-NER activity. Analysis of transcriptional bias and relative mutation rate in intergenic regions of XP-C tumors (Fig. 3e, f) revealed strong dependence on the intensity of XR-seq outside the annotated genic regions. This extended TC-NER activity outside of the transcribed strand of genes is especially strong in early replicating regions with a high density of active genes (Fig. 3c). It may explain the decrease of the mutation density in intergenic regions and on the untranscribed strands of genes in early replicating genomic regions of GG-NER deficient XP-C samples (Fig. 2d, Supplementary Fig. 5).

**XP-E demonstrates reduced GG-NER activity**

The sensors of UV-induced DNA lesions in GG-NER, XPC, and DDB2 (XPE) are thought to work in tandem when DDB2 binds directly to a lesion and facilitates recruitment of XPC, which in turn initializes the repair process with the TFIIH complex[25]. We decided to compare the features of UV-induced mutagenesis in XP-E resulting from the loss of DDB2 with XP-C and sporadic tumors.

MDS plot based on SBS mutational profiles (Fig. 4a) and hierarchical clustering (Supplementary Fig. 8a) revealed three clusters corresponding to XP-C, XP-E, and sporadic tumors. At the same time, the proportion of CC > TT DBS was much increased in XP-C (0.21) versus sporadic cSCC (0.064), but significantly decreased in XP-E cSCC (0.034, $p = 0.0003$; Mann–Whitney $U$ test, two-sided), confirming qualitative differences of mutagenesis in XP-E. Unlike XP-C, the distribution of the mutational load in intergenic and untranscribed strand gene regions by RT in XP-E was very close to that of sporadic cSCC, suggesting that repair in early RT regions was functional in XP-E (Fig. 2d, Supplementary Fig. 5). Similarly, the MDS plot and hierarchical clustering based on the local mutation load in 2684 1Mb-long intervals along the genome, revealed no strong difference between XP-E and sporadic samples while XP-C samples all grouped together irrespectively the tumor type (Fig. 4b, Supplementary Fig. 8b). For example, a single XP-E melanoma sample clustered within sporadic melanomas

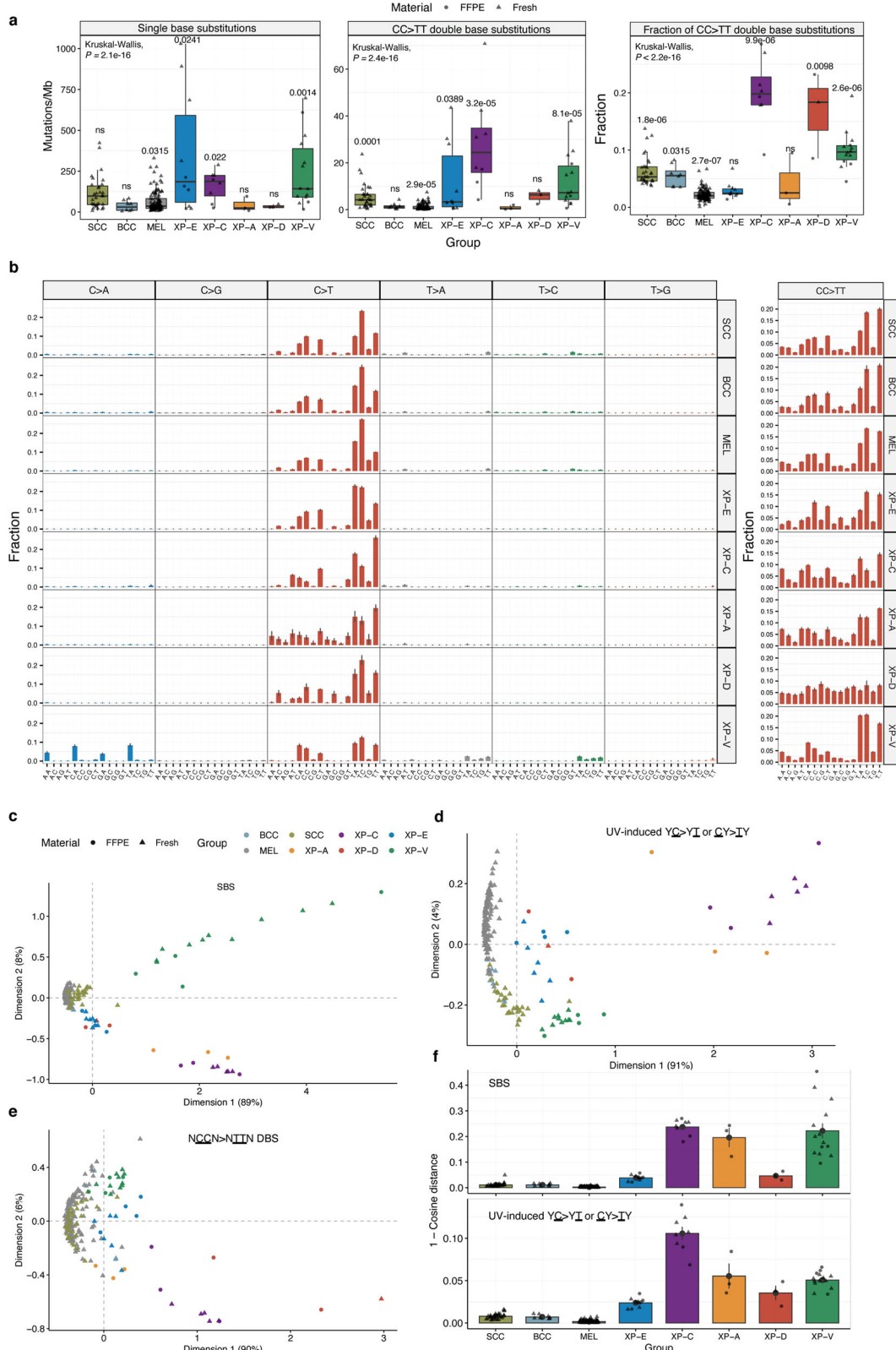

and majority of XP-E nonmelanoma skin cancers—with sporadic samples of the corresponding cancer types, while XP-C samples grouped together separately from sporadic and XP-E tumors (Fig. 4b, Supplementary Fig. 8b).

The XP-E group demonstrated a strong TRB (1.77-fold), which was intermediate between sporadic cSCC (1.33) and the XP-C group (2.47) (Fig. 3a, Fig. 4c, d). Given that TC-NER is functional in XP-E, XP-C, and

sporadic samples, and assuming that GG-NER is fully abrogated in XP-C, we can estimate the relative efficiency of GG-NER in XP-E tumors. Providing all else is equal, GG-NER is 64% less efficient in XP-E than in sporadic cancers.

To provide a more detailed view of the mutation difference between XP-E, XP-C, and sporadic tumors, we compared the association of mutation load in each group with the core epigenetic marks

**Fig. 1 | Mutation landscape of the studied cancers. a** Tumor mutation burden of all single base substitutions (SBS; left panel), double base substitutions (CC > TT; middle panel) per group and a proportion of CC > TT DBS relative to C > T SBS in pyrimidine context (right panel). All the cancer types were combined together per XP group. *P*-values from nonparametric ANOVA are indicated (Kruskal-Wallis test for global *P*-value estimation and Mann–Whitney *U* test, two-sided for individual groups *P*-values). Boxes depict the interquartile range (25–75% percentile), lines– the median, whiskers–1.5× the IQR below the first quartile and above the third quartile. Source data are provided as a Source Data file. **b** Trinucleotide-context mutation profiles of SBS (left panel) and tetranucleotide-context mutation profiles of CC > TT DBS (right panel) per group. Data are presented as mean values +/− SEM. **c** Multidimensional scaling (MDS) plot based on the Cosine similarity distance between the SBS trinucleotide-context mutation profiles of the samples. **d** MDS

plot based on the Cosine similarity distance between the trinucleotide-context mutation profiles of the samples using only C > T mutations with an adjacent pyrimidine (YC > YT or CY > TY), the typical UV mutation context. **e** MDS plot based on the Cosine similarity distance between the tetranucleotide-context mutation profiles of the samples using only CC > TT double base substitutions. **f** Mean Cosine dissimilarity (1-Cosine distance) between original and reconstructed trinucleotide-context mutation profiles using only SBS7a/b/c/d COSMIC mutation signatures for all SBS (upper panel) and C > T mutations with adjacent pyrimidine only (lower panel). Data are presented as mean values +/− SEM. Source data are provided as a Source Data file. Sample size for all the panels (tumors): *n* = 31 for SCC, *n* = 8 for BCC, *n* = 113 for MEL, *n* = 10 for XP-E, *n* = 8 for XP-C, *n* = 3 for XP-A, *n* = 3 for XP-D and *n* = 14 for XP-V.

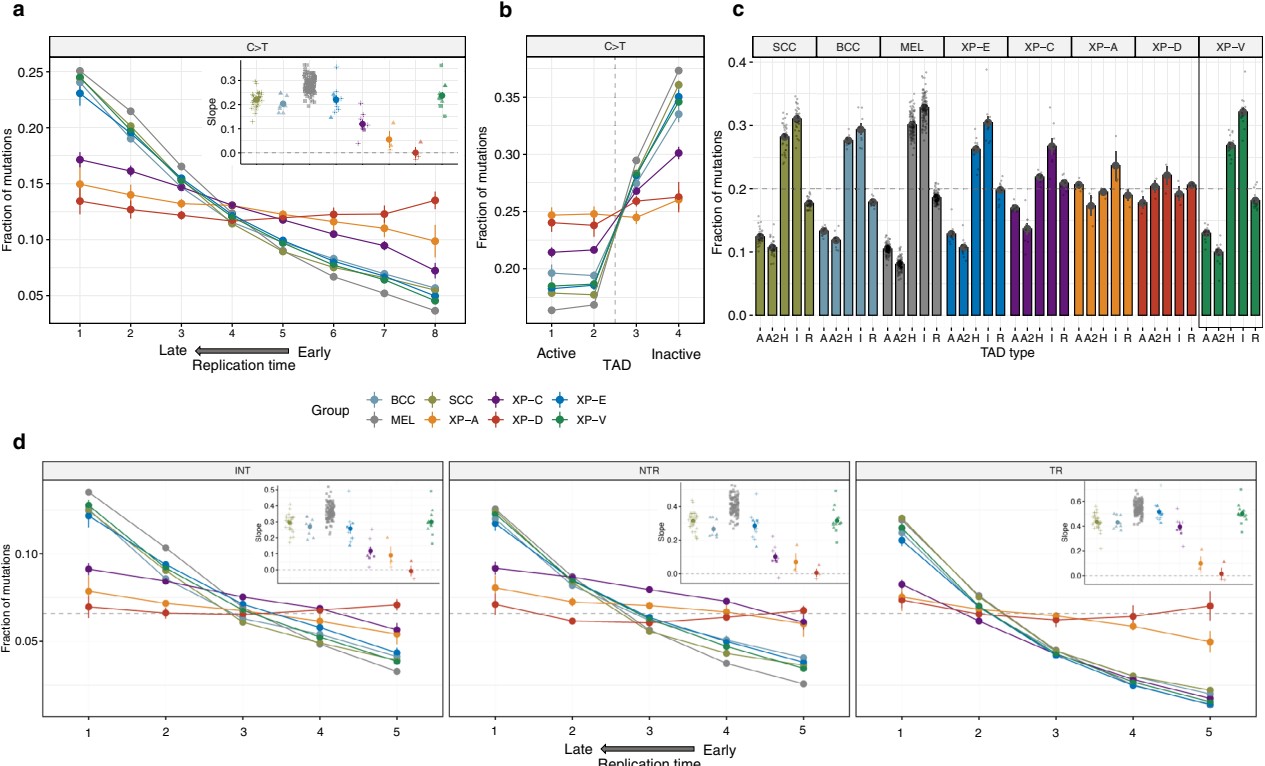

**Fig. 2 | Genomic topography of mutagenesis in the skin cancers (SEM intervals are indicated). a** Fraction of C > T mutations from pyrimidine dimers in genomic regions grouped in 8 equal size bins by replication timing (RT) for XP groups and sporadic skin cancers. The box contains the slope values from linear regressions across 8 RT bins. Data are presented as mean values +/− SEM. Source data are provided as a Source Data file. **b** Fraction of C > T mutations from pyrimidine dimers per group in 1 Mb regions centered at the boundary between active and inactive topologically associated domains (split into two bins each). Data are presented as mean values +/− SEM. Source data are provided as a Source Data file. **c** Fraction of C > T mutations from pyrimidine dimers per group across different chromatin states (R – repressed, A and A2 – active, H – heterochromatin,

I – inactive). Data are presented as mean values +/− SEM. Source data are provided as a Source Data file. **d** Fractions of C > T mutations from pyrimidine dimers in intergenic regions (left panel), on the untranscribed (middle panel) and transcribed (right panel) DNA strands of gene regions grouped in 5 equal size bins by replication timing (RT) for XP groups and sporadic skin cancers. The boxes contain the slope values from linear regressions across 5 RT bins. I intergenic regions, NTR untranscribed strand of genes, TR transcribed strand of genes. Data are presented as mean values +/− SEM. Source data are provided as a Source Data file. Sample size for all the panels (tumors): *n* = 31 for SCC, *n* = 8 for BCC, *n* = 113 for MEL, *n* = 10 for XP-E, *n* = 8 for XP-C, *n* = 3 for XP-A, *n* = 3 for XP-D and *n* = 14 for XP-V.

from primary keratinocyte cell line[26] using only cSCC samples (Supplementary Fig. 9). Unlike XP-C, XP-E tumors did not show strong and significant differences from sporadic cSCC in the dependence of mutagenesis on the majority of epigenetic covariates except for the histone modification marks H3K36me3, H3K27ac and H3K9me3 on the transcribed strand of gene regions (Supplementary Fig. 9).

Taking these observations together, we can speculate that in XP-E tumors, there is a residual activity of GG-NER associated with the ability of XPC to find a fraction of DNA lesions and initiate NER. This correlates with the clinical observation that XP-E patients develop less and later skin tumors than XP-C patients.

## Polymerase η deficiency causes a specific mutation profile in skin cancers

The analysis of XP-V skin cancers revealed that on average of 27% (15–42%) of SBS were represented by C:G > A:T mutations with a highly specific 3-nt context (NCA) and a strong and homogeneous TRB (Figs. 5a, 1b, Supplementary Fig. 1). Similar mutation contexts and a TRB was observed for a part of T:A > A:T mutations, which represented 8.7% of SBS. In sporadic skin cancers, C:G > A:T and T:A > A:T mutations represented only 2.5% and 4.6%, respectively, and had different broad 3-nt contexts without a strong TRB (Fig. 1b, Supplementary Fig. 1). Enrichment of these types of mutations in XP-V suggests that they

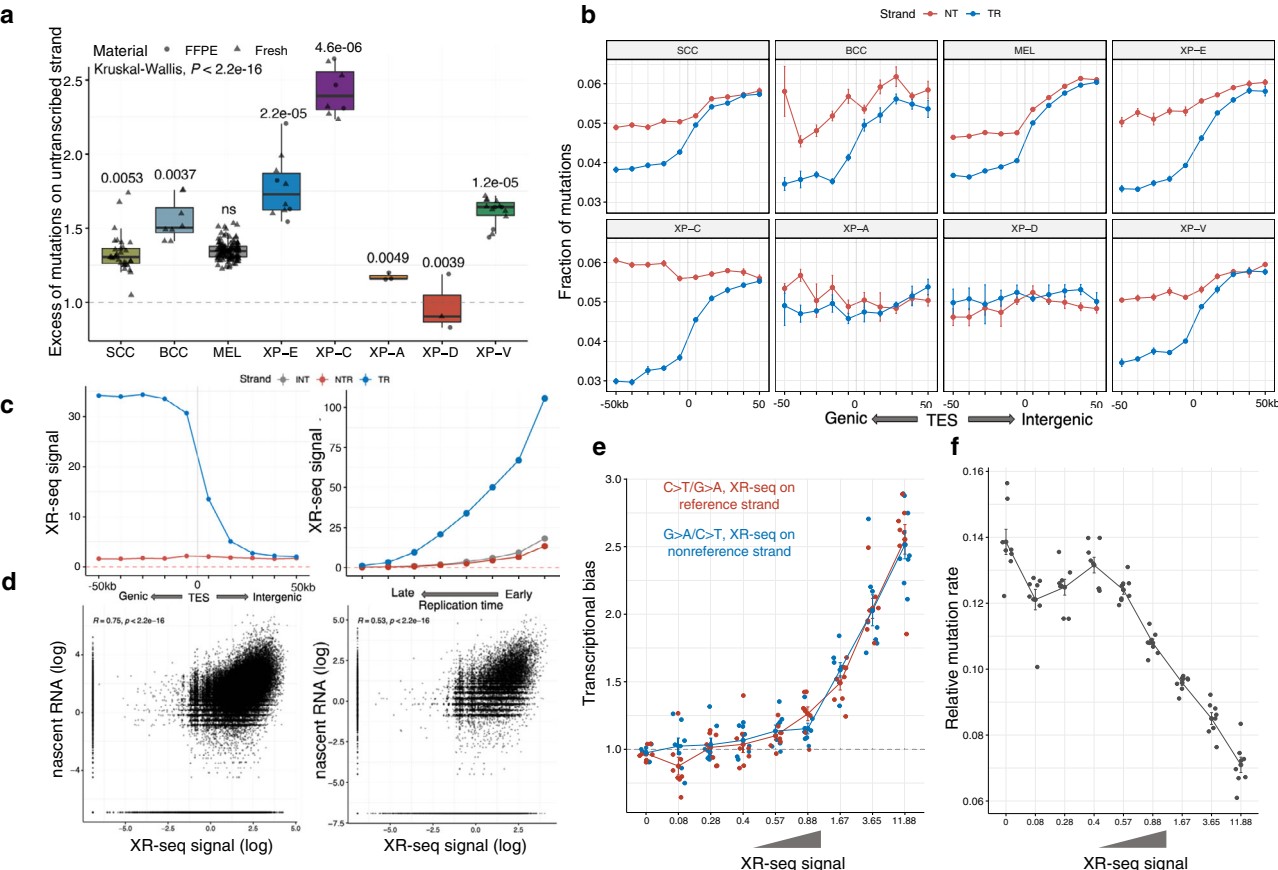

**Fig. 3 | TC-NER activity behind transcription end sites (TES) of genes. a** The transcriptional bias (TRB) per group (ratio between untranscribed and transcribed strand) for C > T mutations with adjacent pyrimidines for XP groups and sporadic skin cancers. *P*-values from nonparametric ANOVA are indicated. Boxes depict the interquartile range (25–75% percentile), lines–the median, whiskers–1.5× the IQR below the first quartile and above the third quartile. $n = 31$ for SCC, $n = 8$ for BCC, $n = 113$ for MEL, $n = 10$ for XP-E, $n = 8$ for XP-C, $n = 3$ for XP-A, $n = 3$ for XP-D and $n = 14$ for XP-V (tumors). Source data are provided as a Source Data file. **b** Fractions of C > T mutations with adjacent pyrimidines separated by strands in the TES-centered 100 kb region (binned by 10 kb intervals). Data are presented as mean values +/− SEM. $n = 31$ for SCC, $n = 8$ for BCC, $n = 113$ for MEL, $n = 10$ for XP-E, $n = 8$ for XP-C, $n = 3$ for XP-A, $n = 3$ for XP-D and $n = 14$ for XP-V (tumors). Source data are provided as a Source Data file. **c** DNA context-normalized XR-seq density from XP-C cell line on untranscribed and transcribed gene strands in the TES-centered 100 kb region (binned by 10 kb intervals; left panel, $n = 1$). DNA context-normalized XR-seq density from XP-C cell line by replication timing for the transcribed and untranscribed DNA strands of genes and intergenic regions. I intergenic regions, NTR untranscribed strand of genes, TR transcribed strand of genes (right panel, $n = 1$). **d** Correlation between XR-seq intensity from XP-C cell line and nascent RNA-seq for genic regions (left panel, $n = 1$) and intergenic regions 50 kb downstream of TES (right panel, $n = 1$). Pearson's $r$ correlation coefficients and $P$ values are indicated. **e** Transcriptional bias of C:G > T:A mutations on intergenic regions of XP-C tumors depending on the XR-seq intensity of XP-C cell line. SEM intervals are indicated, $n = 8$ tumors. **f** Relative mutation rate of C:G > T:A mutations in intergenic regions of XP-C tumors ($n = 8$) depending on the XR-seq intensity in XP-C cell line. Data are presented as mean values +/− SEM.

might originate from lesions that are bypassed by polymerase η in an error-free manner in sporadic skin cancer, but XP-V cells have to use an alternative polymerase(s) to bypass these lesions.

The direction of TRB for these types of mutations indicates a decrease in mutations from lesions involving purines on the transcribed strand (Fig. 5a). Furthermore, comparison of C:G > A:T mutation frequencies on the transcribed and untranscribed strands with the proximal 5′ intergenic regions confirmed that TRB is indeed associated with a decrease of C:G > A:T mutations on the transcribed strand (Fig. 5b). This suggests that mutations occur due to lesions involving purines, which are NER substrates and are effectively repaired by TC-NER on the transcribed strand (Fig. 5b). Interestingly, C:G > A:T mutations had stronger TRB than YC > YT or CY > TY UV-induced mutations in all bins of genes grouped by the expression level (Fig. 5c). This observation might indicate that those lesions produce a smaller helix distortion and are less visible to GG-NER than UV-induced pyrimidine lesions.

C:G > A:T and T:A > A:T mutations occurred in a very specific dinucleotide context, where a purine is always preceded by a thymine base (TA/G > TT), suggesting that causative DNA lesions might be

thymine-purine dimers (Fig. 5d). The number of mutations in a TG context was strongly correlated with the number of mutations in a TA context ($r = 0.98$, Pearson's $r$ correlation coefficient; Supplementary Fig. 10) in our XP-V skin cancer cohort suggesting coordinated mutation processes.

We hypothesized that if TA/G > TT mutations were not directly or indirectly caused by UV-irradiation their abundance would not be correlated with the typical UV-induced (YC > YT or CY > TY) mutations. UV-induced mutations usually accumulate nonlinearly but depend on the UV exposures, while mutations caused by cell physiological processes (such as purine oxidation, cytosine deamination) accumulate more or less linearly with time[27]. We measured a Pearson's $r$ correlation of TG > TT or TA > TT mutations with typical UV-induced (YC > YT or CY > TY) mutations and observed strong correlations in both cases, $r = 0.78$ ($p = 0.001$) and $r = 0.99$ ($p = 1e-10$), respectively (Supplementary Fig. 10).

To further understand the nature of TG > TT and TA > TT mutations we established a *POLH* knockout (KO) of the RPE-1 *TP53*-KO cell line and sequenced whole genomes of the *POLH* wt and

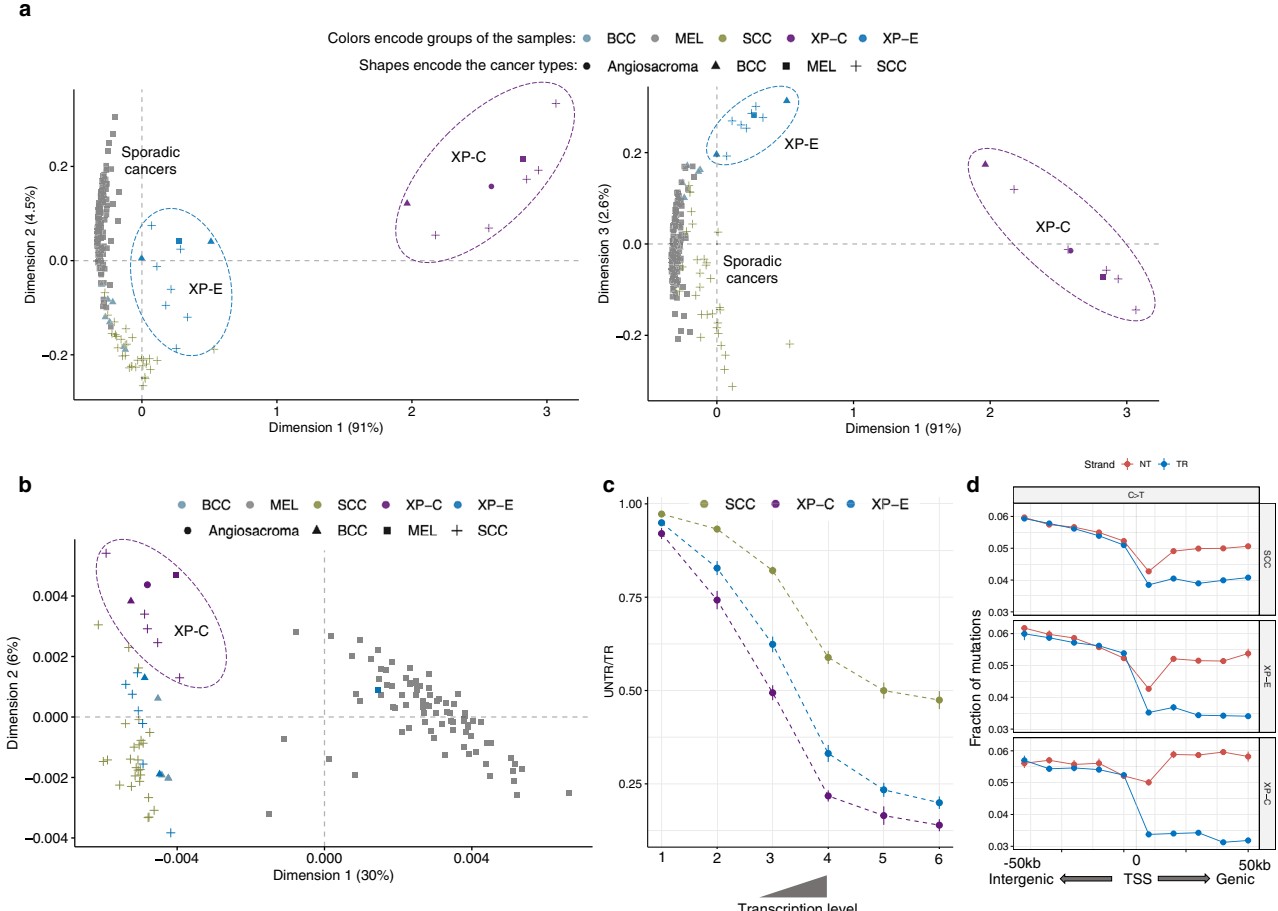

**Fig. 4 | Comparison of genomic mutagenesis between sporadic cancers, XP-E and XP-C groups. a** Multidimensional scaling (MDS) plot based on the Cosine similarity distance between the SBS trinucleotide-context mutation profiles of the samples (Dimensions 1 and 2−left panel, Dimensions 1 and 3−right panel). Colors encode groups of the samples and shapes encode types of cancers. $n = 31$ for SCC, $n = 8$ for BCC, $n = 113$ for MEL, $n = 10$ for XP-E and $n = 8$ for XP-C (tumors). **b** PCA plot based on the density of mutations in 2684 1Mb-long windows along the genome (only for samples with more than 50k mutations belong to sporadic, XP-C and XP-E groups). Colors encode groups of the samples and shapes encode types of cancers. $n = 4$ for BCC, $n = 83$ for MEL, $n = 26$ for SCC, $n = 8$ for XP-C and $n = 10$ for XP-E (tumors). **c** The transcriptional bias (TRB; ratio between untranscribed and

transcribed strand mutation number) for C > T mutations from pyrimidine dimers in genes grouped in 6 bins by gene expression level. Only cutaneous SCC tumors were used for XP-C and XP-E groups. Data are presented as mean values +/− SEM. $n = 31$ for SCC, $n = 5$ for XP-C and $n = 7$ for XP-E (tumors). Source data are provided as a Source Data file. **d** Fractions of C > T mutations from pyrimidine dimers separated by strands in the TSS-centered 100 kb region (binned by 10 kb intervals). Blue line−untranscribed strand for purines or transcribed for pyrimidines, red line −transcribed strand for purines or untranscribed for pyrimidines. Data are presented as mean values +/− SEM. $n = 31$ for SCC, $n = 5$ for XP-C and $n = 7$ for XP-E (tumors). Source data are provided as a Source Data file.

*POLH*-KO clones both without treatment and with treatment with $KbrO_3$ (to induce reactive oxygen species; $n = 1$), UV-A ($n = 3$) and UV-C ($n = 3$) (Supplementary Table 2, Supplementary Fig. 11). There were no major differences in the number of mutations and mutational profiles between *POLH*-wt and *POLH*-KO for untreated cells and $KbrO_3$-treated (Fig. 5e−g). UV-A and UV-C exposures greatly increased number of SBS in the *POLH*-KO cells (3.9 and 10.5-folds respectively, $P = 0.00078$ and $P = 0.01507$, respectively; Welch two sample t-test) and dramatically changed the mutational profiles in comparison with *POLH*-wt clones (Fig. 5e−g). UV-A-treated *POLH*-KO clones had on average 16% of T$\underline{G}$ > TT mutations and 12% of the T$\underline{A}$ > TT mutations with specific to XP-V context and strong transcriptional bias while in the UV-C-treated clone these percentages were 10% and 4% on average, respectively (Fig. 5f, g). UV-treated *POLH*-KO cells demonstrated a distinct pattern of T$\underline{G}$ > VT DBS substitutions (V − A, C or G). Interestingly, a similar DBS pattern was also visible in XP-V tumors (Supplementary Fig. 12).

Another feature of the XP-V skin cancer profile was the presence of 15% (range 11−23%) of mutations originating from TT pyrimidine dimers. Such mutations are very rare in sporadic cancer (4.8%) because

TT pyrimidine dimers are bypassed by polymerase η in a relatively error-free manner. Two predominant types of mutations at TT were T$\underline{T}$ > T$\underline{A}$ and T$\underline{T}$ > T$\underline{C}$, and they, as expected for mutations from pyrimidine lesions, demonstrated strong TRB and were correlated with the typical UV-induced Y$\underline{C}$ > Y$\underline{T}$ or $\underline{C}$Y > $\underline{T}$Y mutations (Fig. 5a, Supplementary Fig. 10). The reconstruction of RPE-1 mutational profiles with the COSMIC mutational signatures revealed significantly higher reconstruction error for *POLH*-KO cells, then for *POLH*-wt cells and overall poor reconstruction performance for UV-A-treated cells (Supplementary Fig. 13) indicating poor representation of inferred mutational process in the public mutational catalogs.

### In the absence of polymerase η, error-prone bypass of 3' nucleotides in pyrimidine dimers shapes the mutation profile of XP-V tumors

The 3-nt context of C > T substitutions in XP-V skin cancers differed from sporadic skin cancers and other XP groups (Fig. 1b, d). Previously it was shown that in the absence of polymerase η, the bypass of CPD photoproducts can be performed in two steps by two TLS polymerases, one of which inserts a first nucleotide opposite to a 3'

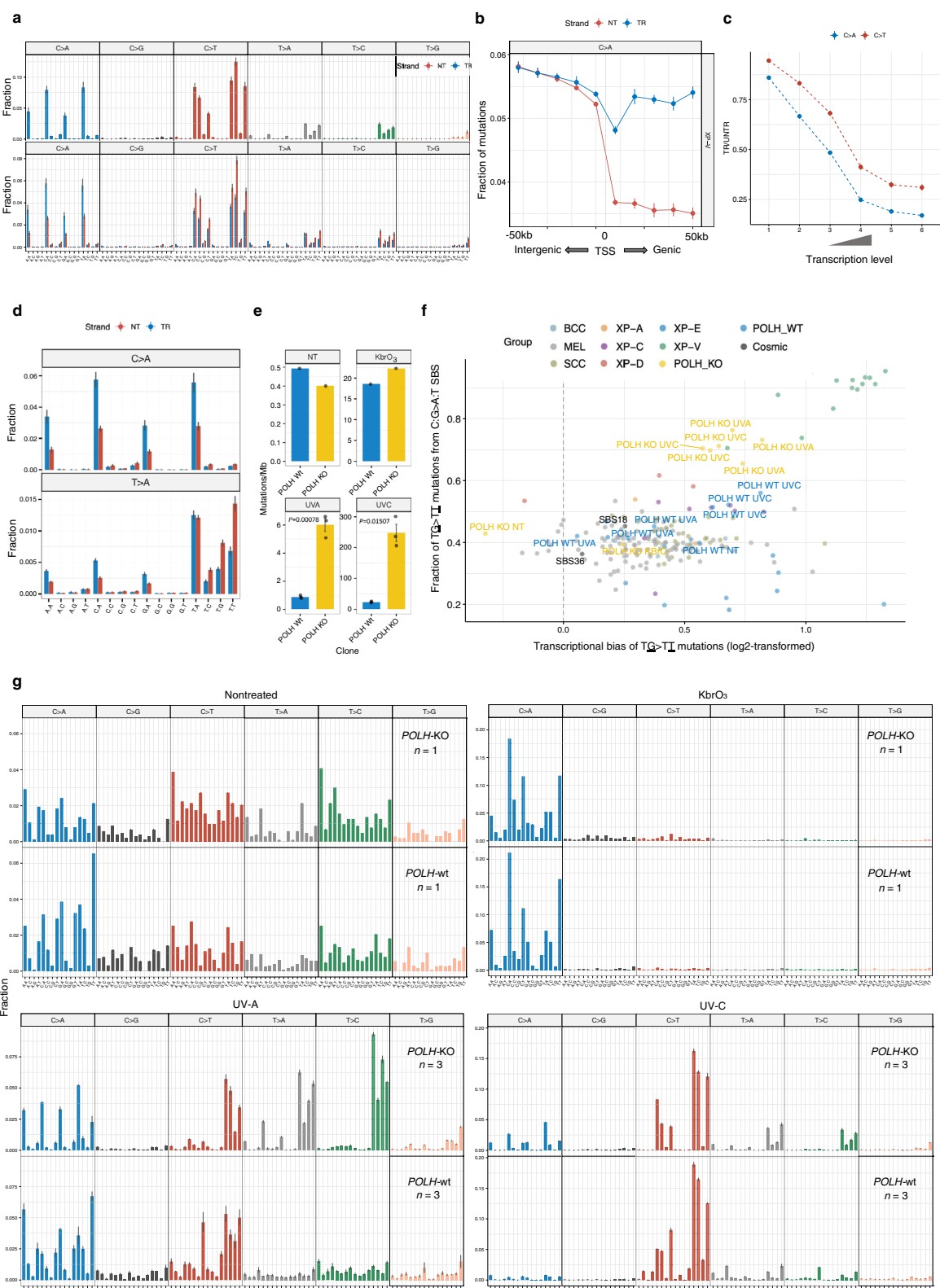

nucleotide of the lesion ("inserter"), and then is replaced by another TLS polymerase, which performs the extension opposite to the 5' nucleotide of the lesion ("extender")[28]. We hypothesized that loss of polymerase η in skin cancer might change the probabilities of mutations at 3' versus 5' nucleotides in pyrimidine dimers and thereafter contribute to the observed differences of the mutation profiles for C > T SBS in XP-V versus sporadic skin cancer.

To test this hypothesis, we first estimated the relative number of mutations arising at 3' and 5' cytosines in the tetranucleotide ACCA, where we could unambiguously allocate a pyrimidine dimer (Fig. 6a). In sporadic skin cancers, the probabilities of mutations at 3' and 5' cytosines were similar, with only a slight increase of mutagenesis from the 3'C (55%), while in XP-V skin cancers 97% of the mutations were from the 3'C (Fig. 6b). This bias towards 3' pyrimidine mutations was

**Fig. 5 | Mutation profiles of XP-V skin cancers and POLH-KO clones.**
**a** Trinucleotide-context mutation profile of genomic SBS (upper panel) and genic SBS (lower panel) separated by transcribed (TR) and untranscribed (NT) strands in XP-V tumors. Blue bars− untranscribed strand for purines or transcribed for pyrimidines, red bars−transcribed strand for purines or untranscribed for pyrimidines. Data are presented as mean values +/− SEM. *n* = 14 tumors. **b** Fractions of C > A mutations separated by gene strands in the TSS-centered 100 kb region of XP-V tumors (binned by 10 kb intervals). Blue−untranscribed strand for mutations from purines and transcribed strand for mutations from pyrimidines; red− transcribed strand for mutations from purines and untranscribed strand for mutations from pyrimidines. Data are presented as mean values +/− SEM. *n* = 14 tumors. **c** The transcriptional bias (ratio between transcribed and untranscribed strand) for C > A and C > T mutations per bin of gene expression level (only XP-V samples represented by BCC, *n* = 11 tumors). Data are presented as mean values +/− SEM. **d** Trinucleotide-context mutation profiles of SBS separated by strands in XP-V tumors for C > A and T > A mutations. Data are presented as mean values +/− SEM,

*n* = 14 tumors. **e** Mutations per megabase in the *POLH* wt and *POLH*-KO clones in nontreated cells (NT, *n* = 1 per cell line independent biological replicate), treated with KbrO₃ (*n* = 1 per cell line independent biological replicate), UV-A (*n* = 3 per cell line independent biological replicates) and UV-C (*n* = 3 independent cell clones per cell line). Welch two sample *t*-test, two-sided. Data are presented as mean values +/− SEM. Source data are provided as a Source Data file. **f** Mutational specificity of the TG > TT mutations in XP-V tumors and *POLH*-KO UV-A- and UV-C-treated cell lines. *X*-axis: log2-transformed transcriptional bias of the TG > TT mutations per genome. *Y*-axis: Fraction of the mutations in the TG > TT context from the total number of C:G > A:T substitutions per genome. *POLH*-KO and *POLH*-wt clones are specifically indicated with their corresponding treatment (KbrO₃, UV-A and UV-C) as well as COSMIC SBS18 and SBS36 mutational signatures associated with oxidative DNA damage (black dots). **g** Mutation profiles of the *POLH*-wt and *POLH*-KO clones for nontreated cells (NT), treated with KbrO₃, UV-A and UV-C. Data are presented as mean values +/− SEM for UV-A and UV-C experiments. Sample size is indicated on the plots (independent cell clones).

also much stronger in XP-V versus other groups of skin cancer for the CT, TC, and TT pyrimidine dimers. For example, ATCA > ATTA mutations were 9.17-fold more frequent than ACTA > ATTA mutations in XP-V than in the other groups (normalized to the corresponding 4-nt frequencies in the human genome). A similar effect was observed for T > A and T > C mutations in ATTA context (Fig. 6c).

These results demonstrate that mutations at pyrimidine dimers in XP-V occur predominantly at the 3' nucleotide, which might be associated with the error-prone activity of the inserter polymerase which replaces polymerase η, and modulate the mutational profile of C > T substitutions. *POLH*-KO cells treated with UV-C conversely demonstrated a very strong bias in CC pyrimidine dimers towards mutations at 3'C (99%) (Fig. 6d).

## Mutation properties of XP groups modulate protein-damaging effects of mutagenesis

High mutation rates in cells increase cancer risk and intensify tumor evolution, while the topography of mutagenesis and mutation signatures can impact the probability of damaging or driver mutations[29–31]. In our dataset of skin cancers, the number of oncogenic mutations in the cancer genome was strongly correlated with the total mutation burden (Fig. 7a).

Active DNA repair in open chromatin regions decreases the accumulation of mutations in the early replicating gene-rich regions of cancer genomes (Fig. 2a). We estimated a fraction of mutations per genome falling in the exonic regions across the studied skin cancer groups and found in XP-A and XP-D tumors a significant enrichment of exonic mutations in comparison with the other groups (Fig. 7b). The effect was caused by the redistribution of mutations from late to early RT regions of a genome (Fig. 2a).

C > T transitions, which are the most prevalent UV mutations, have relatively low protein-damaging effect in the human genome and their damaging/silent mutation ratio is 1.8, while other types of mutations, such as C:G > A:T transversions or CC > TT DBS are more damaging with a damaging/silent mutation ratio of 3.4 and 29.5, respectively (Fig. 7c). Enrichment of highly protein-damaging CC > TT DBSs was particularly pronounced in XP-C and XP-D tumors (Fig. 7c). To better understand how the NER deficiency modulates the protein-damaging effect of UV irradiation we grouped protein-damaging mutations into 5 categories: C > T mutations on the transcribed and untranscribed strand, CC > TT double base substitutions on the transcribed and untranscribed strands, and other SBSs (Fig. 7d, Supplementary Fig. 14). The largest fraction of protein-damaging mutations was accounted for by C > T substitutions in all cancer groups except XP-V where other mutation classes play a more important role. Mutagenesis in splice-sites was preferentially caused by C > T mutations originating from the lesions on the transcribed strands of genes (Supplementary Fig. 14) At the same time different mutation types did

not affect the relative abundance of conservative and nonconservative missense mutations.

Contribution of damaging C > T mutations from transcribed and untranscribed strands of genes (measured as untranscribed/transcribed ratio) differed between groups. It was balanced between strands in sporadic skin cancers (1.02-fold); at the same time the majority of damaging mutations in GG-NER deficient XP-E and XP-C groups were attributed to the untranscribed strand (1.36 and 1.82-fold, respectively), while in GG- and TC- NER deficient XP-D and XP-A groups – to the transcribed strand (0.77-fold and 0.65-fold, respectively, Fig. 7d). These results can be explained by the fact that UV-induced C > T SBS, which originate from the lesions on the transcribed strand, are 1.88-fold more protein-damaging as compared to the untranscribed strand of genes; thereafter, active lesion removal by TC-NER from the transcribed strand of genes results not only in reduction of a total number of mutations from UV lesions, but is particularly important for the reduction of the burden of protein-damaging mutations.

## Discussion

Our results indicate that XP skin cancers deficient in GG-NER (XP-C, XP-E) or polymerase η (XP-V) harbor 3.6-fold more mutations than sporadic skin cancers, on average (Table 2). The mutation profiles in all XP groups were dominated by C > T mutations in pyrimidine dimers; however, they differed from sporadic skin cancers and each other. These differences can be partially explained by the increased contribution to mutagenesis of early replicating GC-rich regions in XP groups with significantly impaired NER (XP-C, XP-A, XP-D). Mutational differences in XP-E might be further explained by the important role of XPE (DDB2) protein in removal of CPD lesions rather than of 6-4PP lesions, which can be recognized by XPC directly[32]. Thus, observed distinct mutational profiles in XP-E and XP-C might be associated with the different relative contributions of CPD and 6-4PP lesions to mutagenesis. Moreover, CC > TT double base substitutions, a characteristic feature of skin cancers, which is particularly enriched in XP-C (but depleted in XP-E), could be associated with 6-4PP photolesions. In addition, CC > TT DBS were not strongly enriched or depleted in XP-V, which might suggest that the occurrence of these mutations does not depend exclusively on polymerase η (Table 2).

The current knowledge about TLS across UV lesions posits that CPD are bypassed by polymerase η alone, while bulkier 6-4PP require two TLS polymerases, one of which performs insertion and the other an extension[33]. The different strategies and TLS polymerases the cells use to bypass CPD and 6-4PP lesions suggest they may result in different mutagenic consequences. While CPDs are probably responsible for most of the mutations detected in XP-V skin tumors, we cannot rule out the possibility that some mutations are due to the replication of 6-4PP by polymerase ζ, θ or even Primpol[34–36]. Several previous

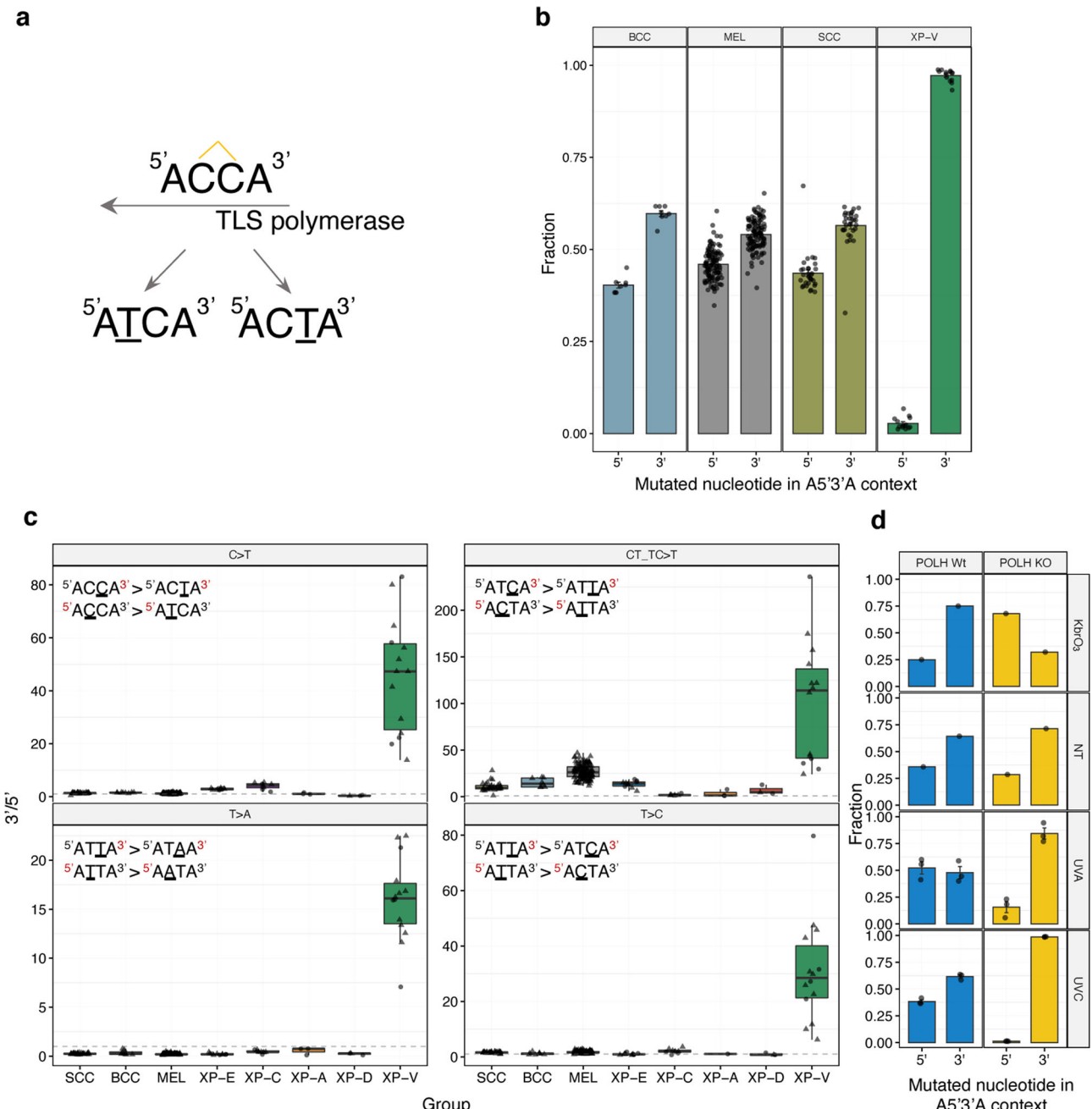

**Fig. 6 | Dimer translesion bias in XP-V skin cancers. a** Schematic representation of the putative CC photodimer in ACCA context and resulting mutations analyzed in the panel **b**. **b** Fraction of C > T mutations from 5' and 3' cytosines of the dimer in the $^{5'}$ACCA$^{3'}$ context per group of tumors. Data are presented as mean values +/− SEM. $n = 31$ for SCC, $n = 8$ for BCC, $n = 113$ for MEL and $n = 14$ for XP-V (tumors). Source data are provided as a Source Data file. **c** "Dimer translesion bias" for different sequence contexts per group of tumors. Comparison of C > T mutation frequency in CT and TC pyrimidine dimers was performed after normalization to the number of such contexts in

the genome (upper right panel). Boxes depict the interquartile range (25–75% percentile), lines - the median, whiskers − 1.5× the IQR below the first quartile and above the third quartile. $n = 31$ for SCC, $n = 8$ for BCC, $n = 113$ for MEL, $n = 10$ for XP-E, $n = 8$ for XP-C, $n = 3$ for XP-A, $n = 3$ for XP-D and $n = 14$ for XP-V (tumors). Source data are provided as a Source Data file. **d** Fraction of C > T mutations from 5' and 3' cytosines of the dimer in the $^{5'}$ACCA$^{3'}$ context in the RPE-1 *POLH*-wt and *POLH*-KO clones. SEM intervals are indicated. $n = 1$ for NT and KbrO$_3$ and $n = 3$ for UV-A and UV-C clones per cell line. Source data are provided as a Source Data file.

works proposed that in the absence of polymerase η backup TLS polymerases, such as polymerases ι or κ, insert a nucleotide opposite to the 3' base of a CPD with high error rate[37–40]. The extension could be performed by polymerase κ or ζ. In XP-V cancer genomes, we demonstrated a strong increase in the mutagenicity of the 3' nucleotide of the pyrimidine dimer, a phenomenon observed before using lacZ mutational reporter gene in polymerase η – deficient mice[41]. This might be explained by the model where a CPD in the absence of

polymerase η is bypassed in two steps instead of one, by an error-prone inserter polymerase followed by an error-free extender polymerase. We propose to name the effect of differential mutagenicity of 3' and 5' nucleotides in the intrastrand crosslinked DNA dimers as "dimer translesion bias". In this study, we presented an illustrative example of XP-V, where a dimer translesion bias significantly alters the relatively conserved UV-induced mutation profiles for C > T mutations and drives the mutator phenotype of XP-V tumors and *POLH*-KO cell

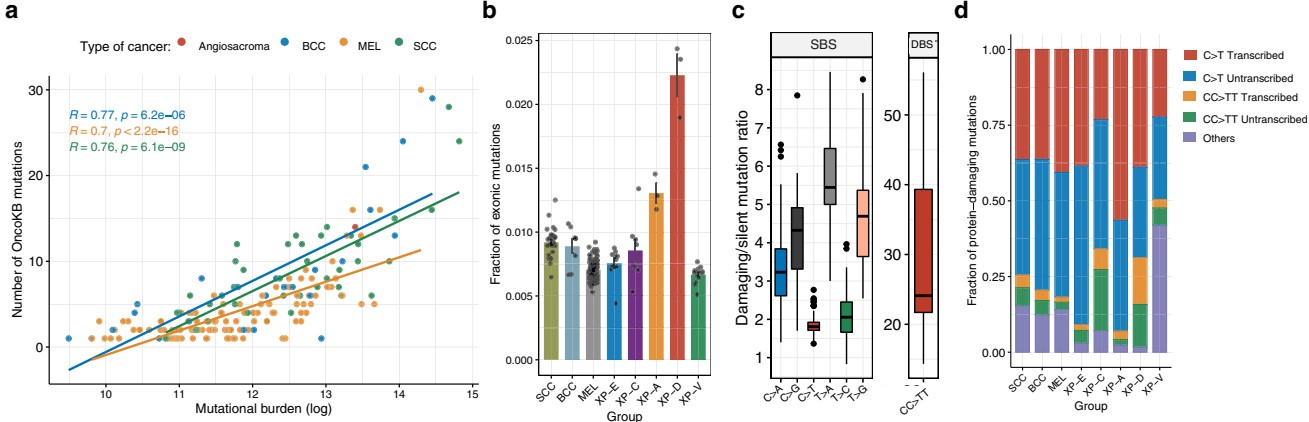

**Fig. 7 | Protein-damaging effect of mutation contexts. a** Correlations between tumor mutation burden and number of oncogenic and likely oncogenic mutations in the studied skin cancer samples according to the OncoKB database. Pearson's *r* coefficients and *P* values are indicated. **b** Mean fraction of exonic mutations from all the mutations per sample. Data are presented as mean values +/− SEM. *n* = 31 for SCC, *n* = 8 for BCC, *n* = 113 for MEL, *n* = 10 for XP-E, *n* = 8 for XP-C, *n* = 3 for XP-A, *n* = 3 for XP-D and *n* = 14 for XP-V (tumors). Source data are provided as a Source Data file. **c** Protein-damaging/silent mutation ratio per substitution type in our pooled skin cancer cohort (*n* = 190 tumors). Damaging mutations—all non-silent exonic (missense, truncating) and splice site mutations. Boxes depict the interquartile range (25–75% percentile), lines—the median, whiskers—1.5× the IQR below the first quartile and above the third quartile. **d** Mean fraction of protein-damaging mutations originating from the main mutation classes split by gene strand per group.

**Table 2 | Schematic summary of the main results from genomic analysis**

| Genomic feature | Sporadic skin cancers | XP-E | XP-C | XP-A | XP-D | XP-V |
|---|---|---|---|---|---|---|
| Mean SBS per MB | 72 | **350** | **162** | 43 | 35 | **248** |
| DBS, % | 4.7 | 3 | **20** | 4 | **17** | 10 |
| Mutation rate heterogeneity across the genome | High | High | Moderate | **Low** | **Low** | High |
| Transcriptional bias | 1.4 | **1.8** | **2.4** | 1.2 | 1 | 1.6 |
| G:C > T:A SBS, % | 3.3 | 0.7 | 3.3 | 1.8 | 1.2 | **27** |
| T > A/C/G SBS in TT context, % | 4.7 | 0.9 | 2.9 | 0.8 | 0.7 | **15** |
| Dimer translesion bias, (5'/3' ratio) | 1.3 | 2.9 | 4 | 1.1 | 0.4 | **45.6** |
| TC-NER activity | High | High | High | Absent | Absent | High |
| GG-NER activity | High | Low | Absent | Absent | Absent | High |

Bold text indicates the most extreme values.

line. An alternative explanation, independent of the backup polymerases involved, could be related to the high deamination rate of cytosines within CPDs. A new approach to map genome-wide cytosine deamination within CPDs recently established that the 3' cytosine is three-fold more prone to deamination when compared to the 5' cytosine[42]. We rejected this hypothesis based on the following results of this study: (i) relative rate of 3' to 5' cytosine deamination events and resulting mutations are expected to be constant for both sporadic and XP-V samples, but we observed strong effect only in XP-V tumors; (ii) the average difference between mutation rates from 3' and 5' cytosines in XP-V samples was 45.6-fold which exceeds all the estimations of differential cytosine deamination, and finally (iii) the same effect was observed for T > A and T > C mutations from ATTA nucleotide context (Fig. 6c), while thymines are not subjected to deamination.

It is well known that the local mutation rate in cancers and in the germline strongly correlates with epigenetic features of the genomic regions[43,44]. The most strong associations were observed for replication timing, chromatin accessibility, active and non-active topologically associated domains, and some chromatin marks such as H3K9me3, H3K27me3, and H3K9me2[44]. Although the chromosomal regions can harbor multiple annotations correlated with each other (early replication timing is associated with active TAD domains and open chromatin regions), they can provide additional information on the biological mechanisms of mutagenesis[43]. In NER-deficient XP-A and XP-D tumors, we observed weak heterogeneity of the mutation frequency across the genome depending on the replication timing, borders between active and inactive TAD domains and chromatin status. The similar but weaker effect was observed for intergenic regions and untranscribed strands of genes in XP-C samples with completely dysfunctional GG-NER, but active TC-NER. This finding demonstrates that the decreased mutagenesis in sporadic skin cancers in large open chromatin regions is driven by their accessibility to NER.

We observed that intergenic genomic regions and untranscribed strands of genes in GG-NER-deficient (but TC-NER-proficient) XP-C samples had a decreased mutation load in early-replicating genomic regions, which are enriched in genes with high levels of transcription. We have shown that it can partially be explained by the activity of TC-NER detected up to 40 Kb on average across genes beyond the annotated TES and propose to name this phenomenon "extended TC-NER". In the early-replicating genomic regions, genes are densely located and transcribed colinearly or in opposite orientations. Extended TC-NER can contribute substantially to the DNA repair independently of GG-NER, thus lowering mutation load in intergenic regions and on the untranscribed strands of closely located or overlapping genes. Early-replicating genomic regions with open chromatin are usually better repaired by GG-NER[12] than late-replicating regions with

compacted chromatin. According to our results extended TC-NER is expected to further lower the mutation rate in euchromatic regions. Future studies exploring XR-seq and nascent RNA-seq data after genotoxic stress in different cell line models including XPC-deficient will be required to better understand the details and associations of the extended TC-NER with the level of gene expression, gene length and particular genotoxin (ex. UV, cisplatin).

Properties of C:G > A:T and T:A > A:T mutations, characteristic of XP-V skin tumors, such as the specific dinucleotide context (TA/G), strong transcription bias, and correlation with the number of C > T UV-induced mutations, enabled us to speculate that these mutations are associated with lesions that directly or indirectly induced by UV. C:G > A:T mutations are unlikely to be associated with 7,8-dihydro-8-oxoguanine (8oxoG) DNA adducts as mutation signatures of 8oxoG (COSMIC signatures SBS18 and SBS36) have different 3-nt context (lack of 5′ thymine specificity) and do not demonstrate transcriptional bias (https://cancer.sanger.ac.uk/signatures/sbs/). Furthermore, rare photolesions in a TA context have been previously reported, and their chemical structure was described as Thymidylyl-(3′−5′)-Deoxyadenosine "TA photoproducts"[45–48]. More recently, a study dedicated to the discovery of atypical photolesions reported rare mutagenic TA photoproducts and, to a lesser extent, TG photoproducts as the most associated with UV-irradiation after CPD and 6-4PP photolesions[49]. The high mutagenesis in TA/G contexts in XP-V tumors uncovers a critical and non-redundant function of polymerase η in the error-free bypass of highly mutagenic but poorly studied DNA lesions, probably induced by UV. The WGS analysis of the RPE-1 *POLH*-KO clones confirmed that TG > TT and TA > TT mutations are greatly increased after UV-A and UV-C exposure, and have shown that they are most prevalent after UV-A. However, after KbrO$_3$ treatment which induces reactive oxygen species, in *POLH*-KO cells SBS mutational pattern did not change as compared to the wild type. Presence of TG > VA DBS in XP-V tumors and in *POLH*-KO cells after UV-exposures further suggests that their origin might depend on thymine-guanine dimers.

Another important peculiarity of the XP-V mutation profile is a high fraction of mutations (on average 15%) originating from a TT dinucleotide (TT > TA/C). It is known that the majority of CPD lesions occur in a TT context[50] and polymerase η is the main polymerase to bypass them in an error-free manner. In the absence of polymerase η, other TLS polymerases perform bypass of TT lesions introducing more errors. Even though TT CPD represents near 50% of UV-induced lesions, the proportion of TT > TA/C mutations is only 15% in XP-V. This means that even in the absence of polymerase η, TT CPD is not a highly mutagenic lesion, probably because other replacing TLS polymerases insert predominantly adenines opposite to the lesion following the "A rule"[51,52]. Interestingly, UV-A treated *POLH*-KO cells harbored 51% of T > A and T > C mutations, while in case of treatment with UV-C, it was only 22% (Fig. 5g). As UV-A produces predominantly CPD photolesions, majority of which are in TT dinucleotides, we speculate that in *POLH*-KO cells treated with UV-A we observed increased contribution of CPDs to mutagenesis. In contrast, UV-C induces many 6-4PPs occurring frequently at TC dinucleotides[53,54]. High yield of 6-4PPs at UV-C exposure might be particularly mutagenic for cytosine-containing pyrimidine dinucleotides in XP-V tumors and *POLH*-KO cells.

In XP skin cancers, UV irradiation results in different mutation profiles and topography of mutagenesis, which are associated with a variation in the probability of protein-damaging and oncogenic mutations. We revealed three main factors contributing to heterogeneity in the proportion of protein-damaging mutations in XP skin cancers, which were associated with the differences in mutation profiles: fractions of transversions and highly damaging CC > TT DBS, the activity of TC-NER, and mutation distribution between open and closed chromatin.

As part of our collection was represented by FFPE-derived samples and in some cases more than one tumor was collected from one patient, we assessed the potential bias associated with these factors on the main analyses in this study. Firstly, hierarchical clustering based on the mutational profiles did not reveal any artifactual grouping of FFPE-derived or same patient derived samples (Supplementary Figs. 3, 15). Secondly, comparison of TMB, TRB, and RT-slopes between FFPE and non-FFPE samples did not reveal any significant differences per group (Supplementary Fig. 16). However, FFPE derived samples in each group demonstrated a tendency towards decreased TMB. Thirdly, we tested if exclusion of single patient's tumors, affected the stability of TMB, TRB and RT-slopes estimations per group and did not observe any strong effects (Supplementary Fig. 17).

The observed differences in mutation burden and mutation profiles might also reflect the differences in clinical manifestations between XP groups. XP-A and XP-D patients show severe sunburn reactions and are diagnosed early. Thereafter, those patients are rarely exposed to UV and have rather few tumors. This might partly explain rather low TMB in XP-A and XP-D skin tumors. At the same time genetic analyses of XP-A skin tumors might be biased due to the fact that all tumors originate from a single patient and that all DNA samples are extracted from FFPE. In tumor-prone XP-C, XP-E, and XP-V groups, the amount and mode of exposure to UV again may be different. XP-C patients do not experience sunburns but develop other skin symptoms resulting in an early diagnosis and sun protection. XP-E and XP-V patients, on the contrary, do not have any symptoms till their 20s or 30s, by which time they may have had a lot of sun exposure, and in subsequent decades, they develop many skin tumors. This might be in line with the observation that mutational profiles in XP-E and XP-V in some features resemble sporadic cancers, while XP-C is the most different.

Overall, our analysis of rare skin cancers with deficient NER or translesion DNA synthesis has revealed how the absence of different NER components modulates mutation burden, profiles, and topography of mutagenesis after UV irradiation (Table 2). We have attempted to provide mechanistic explanations for the mutation consequences of DDB2 (XPE) loss in the XP-E group and the polymerase η deficiency in the XP-V group for UV mutagenesis in skin cancer. Further mutation studies on experimental cell lines from XP patients can extend our knowledge of the role of major and rare photoproducts in skin cancer pathogenesis and biological mechanisms supporting genome stability.

## Methods
### Studied samples
The samples were collected from patients with a confirmed XP diagnosis. Informed signed consents were obtained from patients and/or their parents per the Declaration of Helsinki and the French law. This study was approved by the French Agency of Biomedicine (Paris, France), the Ethics Committee from the CPP of the University Hospital of Bordeaux (Bordeaux, France), the Institutional Review Board of Gustave Roussy (CSET: 2018-2820; Gustave Roussy, Villejuif, France), the Research Ethics Committee of Guy's and St Thomas' Foundation Trust, London (reference 12/LO/0325), and the CONEP (Brazil), Number CAAE 48347515.3.0000.5467. Sex and gender were not considered in the study design and during analysis since the focus of the work was on the mutational profiles of skin cancers.

The tumor samples were collected from patients during surgery. The tumors were stored in liquid nitrogen or allprotect tissue reagent, and 16 in FFPE. Normal control samples were represented by blood (4 patients), saliva (7 patients), fresh skin (2 patients), or FFPE (6 patients). DNA from non-FFPE tissues was extracted using AllPrep DNA/RNA/miRNA Universal Kit (Cat. No. / ID: 80224, Qiagen) according to the manufacturer's instructions. DNA from FFPE blocks was extracted after examination and dissection by a pathologist. Tumor DNA was

extracted from parts of FFPE containing a high fraction of tumor cells using Maxwell® RSC DNA FFPE Kit (Catalog number: AS1450, Promega) according to the manufacturer's instructions. Non-tumoral DNA was extracted from FFPE blocks that did not contain tumor cells if available, or from parts of tumor cell-containing FFPE blocks free from tumor cells. DNA quantity and quality were assessed using the NanoDrop-ND-1000 (Nanodrop Technologies).

## Genome sequencing and variant calling

The genomes were sequenced using BGISEQ-500 in BGI (Shenzhen) according to the manufacturer's protocols to the mean coverage after deduplication equal to 40X for tumor and 30X for normal DNA (100 bp paired-end reads). Reads were mapped using BWA-MEM[55,56] (v0.7.12) software to the GRCh37 human reference genome, and then we used the standard GATK best practice pipeline[57] to process the samples and call somatic and germline genetic variants. PCR duplicates were removed, and the base quality score recalibrated using GATK[58] (v4.0.10.1), MarkDuplicates, and BaseRecalibrator modules. Somatic variants were called and filtered using GATK tools Mutect2, FilterMutectCalls, and FilterByOrientationBias and annotated with oncotator[59] (v1.9.9.0). SCNAs calling was done with FACETS[60] (v 0.5.14). Quality controls of FASTQ files and mapping were done with FASTQC[61] (v0.11.7), samtools[62,63] (v1.9), GATK Hsmetrics, and MultiQC[64] (v1.5). All processing steps were combined in a pipeline built with snakemake[65] (v5.4.0).

## Filtration of somatic variants in tumor samples

Only PASS-filtered somatic variants supported by at least one read from each strand and at least three reads in total with variant allele frequency higher than 0.05 and POPAF filter > 5 (negative log 10 population allele frequencies of alt alleles; probability of the mutation to be a germline polymorphism) were used for the analysis. In addition, all used VCF files were filtered based on the alignability map of the human genome from the UCSC browser (https://genome.ucsc.edu/cgi-bin/hgFileUi?db=hg19&g=wgEncodeMapability) with the length of K-mer equal to 75 bp (wgEncodeCrgMapabilityAlign75mer, mutations overlapped regions with score <1 were filtered out) and UCSC Browser blacklisted regions (Duke and DAC).

To filter out the FFPE artefacts, we employed Support Vector Machine-based (SVM) methodology with the e1071 R library[66]. For each sample separately, each variant in the prefiltered VCF file (the same filters as for the fresh non-FFPE samples) was annotated with additional quality information specific for the alternative allele from the BAM file using bam-readcount utility[67]. This additional BAM-derived information in the form of a table was merged with the quality annotations from the VCF file (VCF was parsed into a table with vcf2tsv from vcflib library[68]) which included CONTQ (Phred-scaled qualities that alt allele are not due to contamination), SEQQ (Phred-scaled quality that alt alleles are not sequencing errors), STRANDQ (Phred-scaled quality of strand bias artifact), TLOD (Log 10 likelihood ratio score of variant existing versus not existing). The typically UV-induced double base substitutions (CC:GG > TT:AA) were considered true positive variants, while abundant FFPE artefacts TG:CA > CA:TG were considered false positive variants during the training of SVM. To tune the SVM parameters we subset 25% of the TG:CA > CA:TG and CC:GG > TT:AA variants and run tune() command (cost = c(0.001,0.01,0.1, 1,5,10,100)). Then the best tuning parameters for the model were chosen (tune.out$best.model) and applied to the training dataset of 50% of the TG:CA > CA:TG and CC:GG > TT:AA variants using svm() command with 10 k-fold cross validations (cross = 10) and probabilistic assignment of the classification (type = "C-classification", probability = TRUE, scale=T) to build the SVM classification model. Finally, the SVM classification model was applied to the whole dataset of variants to classify them as true positive in a probabilistic manner (command predict(), probability = TRUE). We extracted for the downstream analysis only the variants with a probability of being true positive > 0.95.

## Mutation spectrum, MDS, and comparison with known signatures

To convert the VCF files into a catalog of mutation matrices, we used SigProfilerMatrixGenerator v.1.0 software[69]. Before the profiling, VCF files were split into separate files with single base substitutions and other variants to avoid splitting double base substitutions into single base substitutions by the software. To construct the multidimensional scaling plots (MDS), we computed pairwise Cosine similarity distance between all the samples using MutationalPatterns R package[70,71] (cos_sim_matrix()) and then processed the matrix of distances between the samples in the prcomp() function in R. The hierarchical clustering was performed based on the Cosine similarity distances between mutational profiles with cluster_signatures() function of MutationalPatterns R package (method "average"). For the hierarchical clustering visualization, we used Pheatmap R package.

To understand whether known UV signatures can explain the mutational profiles of XP and sporadic datasets, we extracted four SBS mutational signatures previously associated with UV irradiation (SBS 7a,b,c,d) from the COSMIC database[72] (V3.2, https://cancer.sanger.ac.uk/signatures/sbs/) and then reconstructed observed mutational profiles of the studied samples using these four UV-associated mutational signatures (fit_to_signatures(), MutationalPatterns R package). The Cosine dissimilarity of the observed and reconstructed mutational profiles was calculated for each sample as 1-Cosine distance. The procedure was performed separately for all 96 trinucleotide mutational contexts and only 12 mutational contexts of the UV-induced spectra (NCY > NTY or YCN > YTN).

## Replication timing, TADs, epigenetic marks, and mutational load along the genome

We used Repli-Seq data from 11 cell lines[73] (BG02, BJ, GM0699, HeLa, HEPG2, HUVEC, IMR90, K562, MCF7, NHEK, SK-N-SH) to identify conservative replication timing regions. For each 1-kb region, we calculated weighted mean replication timing and then its standard deviation between all the cell lines and removed all the regions with a standard deviation higher than 15. For the rest of consistent regions across different cell lines, we calculated the mean values and used them during analysis. The genome was divided into five or eight bins according to the replication timing values, and mutation density was calculated for each bin, adjusting for trinucleotide contexts. In addition, we computed the dependence of mutation density on replication timing separately for intergenic and genic regions (splitting mutations on the transcribed and untranscribed strands).

The genomic location of the 1MB borders between topologically associated domains (TADs) was downloaded from the recent publication exploring mutation rate dependency on TAD structures[19]. The border regions were spitted into 1-kb intervals and separated into four bins (two for active and two for inactive TADs). Then the fraction of mutations per each sample fallen into each bin was calculated, adjusting for the trinucleotide composition. A similar procedure was performed for the consensus chromatin states of the genome from the same publication.

To calculate the slopes of the mutation load over replication timing (or other epigenetic marks) bins per sample, the logarithm of the normalized fraction of mutations in each bin was fitted into a linear model (lm()) with the number of each bin (1–8).

To investigate the relationships between mutation density and intensity of various epigenetic marks (Dnase, H3K36me3, H3K27ac, H3K4me1, H3K27me3, H3K9me3, methylation level from whole genome bisulfite sequencing), we downloaded bigwig files of the Roadmap Epigenomics Project[26] and converted them to wig and then bed

files (tissue E058, keratinocyte). The mean intensity of each mark was calculated for 1-kb non-overlapping windows across autosomes with BEDOPS v2.4.37 (bedmap) software[74]. The mark intensities were normalized to the 1–100 range, and we used only genomic windows with high alignability (equal to 1) along at least 90% of a window. For each window, we split mark intensities into 5 bins (cut2() function in R) and calculated the trinucleotide-adjusted fraction of mutations per sample per bin for each mark separately for intergenic regions, transcribed and untranscribed strands of genes.

To assess the mutation load distribution along the genome between groups of samples and irrespective of the epigenetic features, we split the genome into 1MB-long nonoverlapped intervals and excluded all the intervals with a mappability score less than 1 over 80% of the interval. For the resulting dataset of 2684 intervals, we calculated the mutation density of C > T substitutions in each interval per sample (with at least 50,000 mutations) and then normalized the mutation density. Finally, the principal component analysis was performed on the resulting matrix.

### Transcriptional bias and XR-seq
Transcriptional strand bias (TRB) was quantified for each sample based on the stranded mutation matrixes generated by SigProfilerMatrixGenerator[69]. We computed inequality between mutations from pyrimidines (C > A/T/G; T > A/C/G) to mutations from purines (G > A/C/T; A > C/G/T) for genes located on the sense and antisense strands of DNA relative to the reference human genome.

To compute TRB between genes expressed with different levels, we used RPKM values of RNA-seq from Epigenetic Roadmap Project[26] represented by keratinocytes (E058) and only samples represented by BCC and cSCC. For each gene, mutations were separated as located on transcribed or untranscribed strands, and genes were divided into six bins by the level of expression.

Following the hypothesis that cytosine-containing DNA lesions caused the majority of mutations, we were also able to compute strand-specific mutation densities around transcription end sites (TESs), and transcription start sites (TSSs). Transcribed and untranscribed strands of genes and adjacent to TES/TSS intergenic regions were treated separately. TESs/TSSs of all annotated genes (GENECODE[75] v38) were retrieved using BEDTools v2.30.0[76], and then regions located ±50 kb of TESs/TSSs were split into 1-kb intervals. The 1-kb intervals that overlapped with other intergenic or genic intervals (represented mainly by overlapped or closely located genes) were removed for this analysis, and the rest were aggregated into 10 bins. We then separately calculated the trinucleotide context-adjusted fraction of mutations per bin per sample for transcribed and untranscribed strands.

XR-seq profiles for XP-C cell lines (XP4PA-SV-EB, GM15983) and nascent RNA-seq data from the HeLa cell line were downloaded from the previous works of Hu et al.[22] and Barbieri et al.[24], respectively. The mean intensity of tracks was calculated for binned 1-kb intervals along the genome and ±50 kb around the TESs. In addition, we downloaded and processed the XR-seq dataset of XP-C cell line experiments of Chiou et al.[23]. The FASTQ files were aligned to the human reference genome with Bowtie 2 aligner v2.4.1[77] in −very-sensitive mode and then processed with BEDTools v2.30.0[76] and BEDOPS v2.4.37 to estimate read counts per intervals.

### Dimer translesion bias
To calculate the relative amount of mutations arising from 5′ and 3′ sides of pyrimidine dimers, we extracted mutations from C > T located in the ACCT context, mutations T > C/A located in the ATTA context, and calculated the ratio of such mutations originating from 3′ C/T to 5′C/T separately for each mutation type with the corresponding 4-nucleotide context. In addition, we calculated the ratio between the number of ATCA > ATTA and ACTA > ATTA mutations per sample, adjusting for the different fractions of ATCA and ACTA four-nucleotides.

### Protein-damaging effects of mutagenesis
To assess the protein-damaging effect of different substitutions, we annotated the VCF files using oncotator[59] software and classified exonic mutations into protein-damaging (missense, nonsense, splice-site) and silent. For the C > T and CC > TT mutations, we separated them by strands and calculated the protein-damaging effect separately for the transcribed and untranscribed strands. The number of putative oncogenic drivers per sample was calculated using the OncoKb[78] database (oncogenic and likely oncogenic events).

### Cell culture
RPE-1 *TP53*-KO cell line was obtained as a gift from Dr. Olivier Gavet lab (Institut Gustave Roussy, France). RPE-1 *POLH*-wt and RPE-1 *POLH*-KO cell lines were cultured in DMEM/F-12 (gibco; life technologies, Ref: 11320033) at 37 °C in a humidified atmosphere containing 5% $CO_2$, supplemented with 10% (v/v) fetal bovine serum (FBS; NB-26-00009).

### Generating *POLH*-KO cell line
*POLH*-KO cell lines were obtained from Synthego company (CA, USA). sgRNA was used to generate the *POLH*-KO cell line. Homozygote knock out was verified by sanger sequencing showing 4 nucleotide insertion.

### UV exposure
Cells were irradiated with 10 J/m² UV-C (200–280 nm) or UV-A (320–400 nm) for 4 sequential exposures both for *POLH*-KO and *POLH*-wt cell lines. Irradiation were performed every 4 days.

### KbrO₃ treatment
IC50 values for KbrO₃ was identified as following protocol. 5000 Cells per each well were plated and grown for 24 h in 96-well plates. Cells were treated in serial diluted concentrations of KbrO₃ (500mM–10 μM). Treatment was last for 96 h. After 4 days, density of cells in each well was quantified using Methylene Blue staining. In the first step, cells (wells) were washed with PBS 1×. Then 100 μl absolute methanol is added to each well and plate was incubated for 1 h at room temperature. Then the wells were let to be dried and 100 μl methylene blue solution (concentration 1 g/L) was added to each well and followed by 1-h incubation at room temperature. Following the staining step, wells were rinsed with water for 2 times and then let the wells to dry. Washing step followed by solubilization of stain by adding 200 μl HCL (0.1 N) in each well and incubation at 60 °C for 30 min. In the last step O.D of each well was measured at 630 nm using BMG FLUOstar OPTIMA plate reader.

*POLH*-wt and *POLH*-KO cells were treated for 8 weeks using 300 μM KbrO₃. Treatment was refreshed every 48 h.

### Single-cell cloning and DNA extraction
Single cell sorting was performed by Flow Cytometry Cell Sorting (FACS) in P96-well plate upon the completion of treatment period and reaching the sufficient number of cells. 5–6 h after sorting the wells were monitored to confirm the presence of a single cell in the well. Three clones per condition were randomly selected to pass to P24-well plate to propagate the cells and extract DNA 18–21 days after cell sorting. Genomic DNA was extracted using the Qiamp DNA mini kit (QIAGEN) according to the manufacturer's instructions.

### Sequencing and bioinformatic analysis of the cell line experiments
The RPE-1 clonal cell populations were sequenced in BGI, Shenzhen (mean coverage 13.7x, BGISEQ-500 instrument) and bioinformatically processed in the similar way with tumor samples. In total we sequenced 16 genomes of individual clones (Supplementary Table 2). The nontreated samples were used as "normal" and treated as "tumor" during GATK mutect2 calling of somatic mutations (and vice versa). Then we removed all the nonunique mutations

between the clones (module *bcftools iseq*) as well as supported by less than 3 reads in total and at least one read from each strand. Finally, only strictly clonal mutations with VAF > 0.3 were used for the analysis.

## Reporting summary
Further information on research design is available in the Nature Portfolio Reporting Summary linked to this article.

## Data availability
Experimental WGS FASTQ files from tumors and normal tissue of patients generated in this study and corresponding filtered VCF files have been deposited in the European Genome-phenome Archive (EGA) under accession code EGAS00001006732. The raw FASTQ files and filtered VCF files are available under restricted access due to data privacy laws, and will be made available under approval by the data access committee [https://ega-archive.org/dacs/EGAC00001002945]. Access will be provided within approximately four weeks and be available for six months. Experimental sequencing data (WGS, FASTQ files) from the RPE-1 cell line experiments generated in this study have been deposited to NCBI under accession code PRJNA940340 and corresponding VCF files are available on Mendeley Data server (https://doi.org/10.17632/jkjkpvgxyd.1) [https://data.mendeley.com/datasets/jkjkpvgxyd]. These data are freely available. The previously published melanoma samples from sporadic patients[17] referenced in the study (consensus VCF files with SNVs and INDELs) are available in a public repository from the https://dcc.icgc.org/repositories website. Genomic datasets of XP-C and sporadic cutaneous SCC[12] used in this study is available in the dbGaP database under accession code phs000830.v1.p1, access is restricted and can be granted under approval by the data access committee. Previously published WGS FASTQ datasets of sporadic cutaneous SCC and BCC used in this study and provided by Hartwig Medical Foundation[15] under data request DR-108 are available under restricted access after approval of the data sharing committee [https://www.hartwigmedicalfoundation.nl/en/data/data-acces-request/]. Previously published VCF files from the WGS study of metastatic cutaneous SCC[16] are available under restricted access after approval of the data sharing committee (EGAS00001003370). Previously published XP-C angiosarcoma[13] somatic variants are freely available on Mendeley Data server: https://doi.org/10.17632/7cxt72pckw.1 [https://data.mendeley.com/datasets/7cxt72pckw/1]. COSMIC database (v.3.2) of mutational signatures was used for the comparison of mutational profiles with previously identified mutational signatures (https://cancer.sanger.ac.uk/signatures/). Source data are provided with this paper.

## Code availability
All software used is published and/or in the public domain.

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

## Acknowledgements

S.I.N. was supported by grant Foundation ARC 2017, Foundation
Gustave Roussy, and the The French National Cancer Institute –
RPT21145LLA. P.L.K and S.I.N. were supported by grant from the Foun-
dation ARC-ARCPGA12019120001055_1578 (P.L.K. and S.N.). C.F.M.M.
was supported by Fundação de Amparo a Pesquisa do Estado de São
Paulo, FAPESP, Grant# 2019/19435-4. This work was also supported by
Prism – National Precision Medicine Center in Oncology funded by the
France 2030 program and the French National Research Agency (ANR)
under grant number ANR-18-IBHU-0002. The authors are very thankful
to Xiaole Xu (BGI) for the management of sequencing.

## Author contributions

S.I.N. and A.A.Y. designed the study. A.A.Y. performed the data analysis
and prepared figures. A.A.Y. and S.I.N. drafted the manuscript. A.S., A.L.,
and C.F.M.M. commented on the manuscript. F.R. handled biopsies,
performed QC of the samples and DNA extraction. F.R. performed cell
line experiments. P.L. participated in the DNA extraction and sample
handling. K.G. participated in the data analysis. I.P. and L.P. performed
data preprocessing. J.W. performed data curation and submission to
repositories. T.B.P., C.F.M.M., H.F., A.L., F.M.-P., C.N., C.R., P.L.K., and
A.S. collected the samples.

## Competing interests

The authors declare no competing interests.

## Additional information

**Supplementary information** The online version contains
supplementary material available at

Sergey I. Nikolaev.

**Peer review information** *Nature Communications* thanks Fran Supek,
and the other, anonymous, reviewer(s) for their contribution to the peer
review of this work. A peer review file is available.

