## [Peer review file · Nature Communications]

REVIEWER COMMENTS

Reviewer #1 (Remarks to the Author): Expert in mutagenesis, mutational signatures, DNA damage, and functional genomics

In this manuscript, Yurchenko et al. sequenced tumors from patients with deficiencies in xeroderma pigmentosum genes and performed analyses to show that defects in different XP group genes correlate with differences in patterns of UV-induced mutagenesis. They confirmed that nucleotide excision repair and translesion synthesis are major factors shaping the observed mutational profiles. They also showed that features of mutagenesis in XP-V deficient cancers could be phenocopied by knocking out POLH (translesion DNA polymerase eta) in cell culture.

This work is not without interest, but I have serious reservations about it being acceptable for publication at NComms. Overall, I think the interpretations of the data are mostly sensible, but I have my doubts about the quantity and quality of the data itself.

1) There's not that many cancer samples being analyzed. Further, some samples derive from the same patients, so those samples should not be treated as truly independent. All three XP-A samples came from the same patient. 3 out of 8 XP-C samples came from the same patient. In the XP-E group, 4 samples came from one patient, 3 samples from a second patient, and 2 more samples from a third patient. In the XP-V group, 3 samples came from one patient, and two pairs of samples came from two patients. The authors make general statements about XP group cancers but I'm in doubt about whether they should make such generalizations based on small number of samples with multiplicity from an even smaller number of patients.

2) Related to this, some of the samples had low sequencing coverage (as low as 7x). Even for average coverage in the teens, there will be regions that have significantly lower coverage, so I'm not feeling high confidence about all the data.

3) 16/33 samples sequenced by the authors were FFPE-treated. The procedure for filtering out FFPE artefactual variants is described in the methods. But I didn't see any figures showing validation for the procedure to reassure readers that this procedure worked right. What we got was Figure 1a, which to my eye, looks like maybe the FFPE samples had fewer mutations on average than fresh samples. Surely, there are tradeoffs being made because these kinds of procedures are not necessarily obviously correct or incorrect. But there needs to be a more thorough description of this analysis and its outputs for a reader to judge how reliable and believable it is.

4) The variant calls (in VCF format preferably) should be made available. Also, sequencing reads should be posted to EGA.

5) The color palette and symbology for scatter plots are hard to see and understand clearly. Some colors are too similar to tell apart (MEL vs. SCC, for example). I'm very confused with the information coding of Figure 4a and 4b: some cancers are doubly encoded by shapes and colors, but others are not?? The text says "well delineated clusters" but I'm confused about what I'm supposed to be looking at and seeing. Also, "XP-C formed a separate cluster" but I think there are just free XP-C samples? Is that right? They look like they are at one end of a range of points. Calling that a separate cluster seems rather arbitrary.

6) Figure 1f: The authors report reconstruction errors, implying that there are more DNA damage processes happening than just SBS7 from UV. But it's just left dangling there. The reconstructed profiles should be reported, those could give clues to what other signatures might be present.

7) Figure 3c: Are there supposed to be error bars? Are the error bars all so tiny that they are covered by the data points? I wonder if the claim that TC-NER operates out to 50 kb past the TES is a bit overstated because it looks like the signal is pretty much down to baseline by about 30 kb?

8) Figure 5b: Colors are flipped?

9) Figure 5f: Would including SBS16 and 36 in reconstruction reduce the error from Figure 1f to levels comparable to other cancers?

10) Figure 5e, f, g: Was there just one clone sequenced from each condition? I would be very cautious about over-generalizing from $n = 1$. Summary information on these sequencing samples wasn't mentioned? Should also do reconstruction with COSMIC signatures.

11) Figure 7: There should be further analysis showing breakdown of missense vs. nonsense vs. splice site variants. And also conservative vs. non-conservative substitutions for missense. To get a sense of what fraction of variants are likely to have high impact on protein function.

12) The literature on molecular mechanisms for mutagenic bypass of UV lesions should be discussed and cited more specifically, e.g., roles of pol iota and kappa at CPD, and differing mutagenicity of various 6-4 photoproducts.

Reviewer #2 (Remarks to the Author): Clinical expert in Xeroderma Pigmentosum genetics

In the study “ANALYSIS OF SKIN CANCERS FROM XERODERMA PIGMENTOSUM PATIENTS REVEALS HETEROGENEOUS UV-INDUCED MUTATIONAL PROFILES SHAPED BY DNA REPAIR” the authors assessed the genomes of 38 skin tumors of 24 xeroderma pigmentosum patients and compared them to sporadic skin cancer genomes (152 tumors). The patient cohort included the four most frequent XP complementation groups (XP-A, XP-C, XP-V, XP-D) as well as XP-E. No tumors of the very rare complementation groups XP-B, XP-F, and XP-G were included. The analysis of the tumor genomes from included patients with deficient NER (XP-A, XP-C, XP-D, XP-E) or translesion DNA synthesis (XP-V) revealed how the absence of different NER components impacts mutation burden, mutation profiles, and topography of mutagenesis after UV irradiation. For example a significant increase in mutations was found in cancers from patients with defective GG-NER (XP-C, XP-E) or deficient polymerase eta (XP-V). CC>TT double base substitutions were found to be particularly enriched in XP-C but depleted in XP-E. In XP-V cancer genomes the authors observed that the 3' nucleotide of a pyrimidine dimer is much more frequently mutated than the 5' nucleotide. The authors propose to name this differential mutagenicity of 3' and 5' nucleotides in intrastrand crosslinked DNA dimers “dimer translesion bias”. Furthermore, the authors describe a phenomenon which they propose to call “extended TC-NER”. This describes a decreased mutation load in early-replicating genomic regions which may at least partially be explained by the activity of TC-NER up to 50 Kb beyond the annotated transcriptional end sites. A relatively high number of mutations in a TA/G context were observed in XP-V tumors and confirmed in vitro after UV-A and UV-C exposure of RPE-1 POLH-KO cells.

Overall the manuscript addresses a relevant question, is clearly structured, and very well written. Besides analyzing patients samples some results were also confirmed by in vitro studies using a RPE-1 POLH-KO and a respective wild type cell line. The comparison of different XP cancer genomes provides novel insights on how the absence of different NER components modulates the genome with particular consequences for the patients. The number of assessed tumors is decent considering the rarity of the disease and conclusions drawn from the analyses are supported by the data. Methods are described in sufficient details.

Considering the complexity of different samples, analyses, and results I strongly encourage the authors to provide an overview figure (or a table) that graphically summarizes the (main) results (observed differences in mutation burden, mutation profiles and so on). Such an overview will help the reader to grasp the gist of the manuscript more quickly and provide a reference to quickly retrieve the most important information.

Specific considerations:

Figure 1a: Are in the left panel only C>T SBS represented or all SBS? Please clarify in the figure legend. What kind of post-hoc test was used to generate the indicated p-values? SCC, BCC and MEL represent sporadic cancers and XP-E/C/A/D/V represent results from all tumors of patients in these groups?

Line 116: For consistency authors should call DBS “double base substitutions” instead of “tandem base substitutions” throughout the manuscript.

Line 126: A very similar mutation signature as described here for XP-V and shown in figure 1b has been previously reported for SCC in immunosuppressed patients (Signature 32, DOI: 10.1038/s41467-018-06027-1). This should be mentioned and discussed.

Figure 7d: The figure legend within the figure shows “C>T Transcribed” for both red and orange bars. Based on the text I assume the label for the orange bar is supposed to be “CC>TT Transcribed”. Please correct.

Reviewer #3 (Remarks to the Author): Expert in computational cancer genomics and mutational signatures

In this study, Yurchenko, Rajabi et al have sequenced a number of skin tumors from patients with XP syndromes (with different causal genes) and analyzed mutation patterns therein to provide evidence for various NER/TLS mechanisms gleaned from mutational variation between the genotypes and across the genome. This is a solid, comprehensive genomics study with several findings (some of which are bit speculative perhaps, in terms of mechanisms) of reasonably broad interest to cancer evolution, genome instability, and DNA repair fields. I have some queries for the authors:

1- WGS coverage is moderate, at 40x, and the tumor purity is not great (estimated at 41% overall). Do they see that coverage is very limiting for mutation calling, given the low purity? I.e. do the samples and/or genome regions therein with lower coverage also have lower mutation burdens? Please comment on possible implications of this to conclusions of the study.

2- Gene specific C>T mutation signatures they report „some XP groups demonstrated marked differences for C>T mutations in specific contexts, such as enrichment at TCA in XP-E, TCW in XP-C, or NCY in XP-D“. Are these trends consistent across the (few) tumor samples for each group? Since the groups are small there is a concern that an outlier might create a different mutational signature. It is not immediately clear that Fig 1b or the spectra addresses this (it might be read from Supp Fig 1 with some effort).

3- Lines 132-138: does this analysis, comparing with known COSMIC spectra, not make more sense by reporting results separately for SBS7a versus SBS7b vs 7c vs 7d, rather than as I understood with the mean of the four spectra? It was unclear to me what the interpretation of the current result would be.

4- The RT association analysis considered together with the TAD analysis. These two factors (late/early RT and inactive/active TADs) are highly confounded and thus the two analyses are likely redundant. If they wish to show that TADs are associated with mutation rates independantly of the RT (which I doubt, but am ready to be convinced otherwise) they should condition on one variable while testing association with the other. Ditto for chromatin states.

4b. They observe the difference in effect (loss of RT association in mutation rates) between GG-deficient XP-C (partial effect) and GG+TC deficient XP-A/XP-D (full effect). This is somewhat hard to interpret. Would this difference in RT-association between the GG-deficient and GG+TC deficient still be observed if looking only at the nongenic regions?

5- Lines 186-188. „Since early RT regions are particularly gene-rich²¹, we hypothesized that in GG-NER deficient XP groups, decreased mutation load in early RT intergenic regions might be associated with the TC-NER activity beyond gene boundaries“ This is fine, and the analysis that follows is interesting however I worry about mis-interpreting of results by readers if not given enough context. I suspect that in early-replicating regions, there may be depletion of mutations also in non-transcribed regions (i.e. those far from gene boundaries). This should be commented on, or better yet shown with data.

5b. „significant TRB up to 50kb downstream of the furthest annotated transcriptional end sites (TES)“ Please do not provide „up to“ numbers in the text, but rather a median or mean, and some description of the distribution of the TRB length after the TES.

5c. Considering the XRseq analysis, and how the XRSeq signal extends past the TES of genes and how this mirrors the TRBs (providing additional evidence for the interesting finding that there is post-TES transcription with consequences on mutation rates). Are the lengths of segments of the 2 measurements (TRB, XRseq) correlated across genes? Can they be used to estimate the length of this post-TES protected segment for each gene?

6- The XPE (DDB2) analysis is interesting, using mutational data to support the idea that DDB2 is optional in GG-NER, and estimating the drop in efficiency when there is not any DDB2. However I didn't quite follow what is the value in the analysis „To provide a more detailed view of the mutation difference between XP-E, XP-C, and sporadic tumors, we compared the association of mutation load in each group with the core epigenetic marks...“ that there is difference between XPC and XPE mutants was already established by the 1Mb intervals and the dinucleotide signature, so how does this help further. Are the specific marks mentioned in the text somehow interesting or informative?

7- The section about the „mystery lesion“ that is likely to be copied over error-free by pol eta: this is quite interesting and although speculative, the notion of the thymine-purine dimers is worth mentioning for sure. The one concern is using the correlations between different parts of spectra as evidence „we measured a Pearson correlation of TG > TT or TA > TT mutations with typical UV-induced (YC>YT or CY>TY) mutations and observed strong correlations in both cases, $R=0.78$ ($P = 0.001$) and $R=0.99$ “ – isn't this correlation going to be caused, in part, simply by the different „age“ of each tumor? By default any part of spectrum will correlate to some extent with any other part, and this analysis would need to have controls in place to account for that.

7b. This whole section is a bit dense; I imagine it will be clear to specialists but perhaps less so for the general reader. Consider supplying extra explanations e.g. „Interestingly, UV-A treated POLH-KO cells harbored 52% of T>A and T>C mutations, while in case of treatment with UV-C, it was only 22% (Fig. 5g).“ explain why is this observation interesting. (also other examples above)

8- The tetranucleotide signature analysis of pol eta (XPV) cancers is nice. However making the link between the presented asymmetry within the CC and the inserter/extender mechanism may be premature. Could not there be other (more parsimonious) mechanisms that explain this pattern?

9- The „protein damaging-effects of mutagenesis“ section feels a bit out of place, and also perhaps a bit obvious in parts -- of course C>A transversions are more impactful than C>T transitions. The damaging effects of CC>TT might be (regardless of ratio to silent) not too important in absolute terms because of rarity of dinucleotide substitutions. In this reviewer's opinion, the one interesting part in this section is how strand bias compares to impact. Trimming the rest would help. Also in addition to ref 27 other prior work should be cited here that looked at impact of various mutation processes due to DNA repair activity.

10- Data availability: the somatic mutation calls (not only the raw sequencing data) needs to be made available to the community upon article acceptance.

11- Fonts in figures are too small. Also the use of the „eta“ symbol for shared first authorship is a nice touch but will probably need to be changed.

REVIEWER COMMENTS

Reviewer #1 (Remarks to the Author): Expert in mutagenesis, mutational signatures, DNA damage, and functional genomics

In this manuscript, Yurchenko et al. sequenced tumors from patients with deficiencies in xeroderma pigmentosum genes and performed analyses to show that defects in different XP group genes correlate with differences in patterns of UV-induced mutagenesis. They confirmed that nucleotide excision repair and translesion synthesis are major factors shaping the observed mutational profiles. They also showed that features of mutagenesis in XP-V deficient cancers could be phenocopied by knocking out POLH (translesion DNA polymerase eta) in cell culture.

This work is not without interest, but I have serious reservations about it being acceptable for publication at NComms. Overall, I think the interpretations of the data are mostly sensible, but I have my doubts about the quantity and quality of the data itself.

Response: We thank the reviewer for this encouraging summary of our manuscript results and grateful for multiple suggestions which allowed us significantly improve the manuscript and data representation. Below we provide answers to the specific points and additional analysis of the data quality.

R1Q1 There's not that many cancer samples being analyzed. Further, some samples derive from the same patients, so those samples should not be treated as truly independent. All three XP-A samples came from the same patient. 3 out of 8 XP-C samples came from the same patient. In the XP-E group, 4 samples came from one patient, 3 samples from a second patient, and 2 more samples from a third patient. In the XP-V group, 3 samples came from one patient, and two pairs of samples came from two patients. The authors make general statements about XP group cancers but I'm in doubt about whether they should make such generalizations based on small number of samples with multiplicity from an even smaller number of patients.

Response: We agree that not independent samples may introduce bias into the analysis, but we would like to draw attention of the reviewer to some important considerations : (i) Xeroderma pigmentosum is a very rare disease and we were able to collect the largest existing collection of XP skin tumors suitable for WGS analysis (ii) All XP-C samples are actually coming from different patients (Table 1, Supplementary Table 1) (iii) all the tumors from the same patients are non-clonal and are genetically independent, mostly taken at different time points and from different body sites.

Among the five studied XP groups the samples from the same patients are present for XP-E, XP-A and XP-V. For XP-E and XP-V groups we have a decent sample size: 10 and 14 tumors from 4 and 9 patients respectively and this allow us to better understand the effect of sampling from the same individual. To check if the tumor origin can influence the mutational profiles we made non-supervised hierarchical clustering based on the cosine similarity distance between the tumors and have not identified a tendency to group for tumors from the same individuals inside the groups (Supplementary Fig. 15b). To better investigate potential bias introduced by samples from the same patient we plotted results of different analyses visualizing individual patients with color codes (Supplementary Fig. 15a). We then performed

resampling-based analysis independently for XP-E and XP-V groups where we compared the means of the main features (mean TMB, fraction of double base substitutions, transcriptional bias and replication timing slopes) between original dataset and resampled subset composed each time from one tumor per patient (resample x times, taking each time only n unique samples, where x=number of tumors per group and n=number of patients per group). Resulting distributions resampled data were tight and we did not find statistically significant differences between means of original and resampled datasets (see a figure below). To further check if samples from some individuals can introduce a significant bias into the analysis we generated subgroups of tumors (for XP-V and XP-E separately) excluding each time samples from one individual. Then we compared means of these resampled groups using nonparametric ANOVA (for TMB, fraction of double base substitutions, transcriptional bias and replication timing slopes) and did not identify significant influence of any single-patient derived samples to the analysis (Supplementary Fig. 17).

At the same time, such analyses could not be conducted for XP-A group. Thereafter we added a sentence in the discussion stating the representation of this group might be biased due to the fact that all 3 tumors come from single patient (L508-509). In addition, we added a specific paragraph to discuss the limitations of our study related to this and R1Q3 questions (L494-503).

R1Q2 Related to this, some of the samples had low sequencing coverage (as low as 7x). Even for average coverage in the teens, there will be regions that have significantly lower coverage, so I'm not feeling high confidence about all the data.

Response: We agree that low sequencing coverage can in theory bias the analysis and to assess this we calculated Pearson correlation between the number of mutations and coverage of the tumors per group (see figure below). We did not identify strong significant for the majority of samples. There was some positive correlation between the number of mutations and coverage for XP-C samples, but was completely absent for two other groups with the decent samples size (XP-E and XP-V). We would like to note that we have only one sample with 7x sequencing coverage and this sample is a part of well-sampled XP-E group (n=10 samples in total). The majority of other samples from XP-E and other groups are characterized by moderate coverage (Supplementary Table 1).

R1Q3 16/33 samples sequenced by the authors were FFPE-treated. The procedure for filtering out FFPE artifactual variants is described in the methods. But I didn't see any figures showing validation for the procedure to reassure readers that this procedure worked right. What we got was Figure 1a, which to my eye, looks like maybe the FFPE samples had fewer mutations on average than fresh samples. Surely, there are tradeoffs being made because these kinds of procedures are not necessarily obviously correct or incorrect. But there needs to be a more thorough description of this analysis and its outputs for a reader to judge how reliable and believable it is.

Response: We agree that FFPE-derived DNA as a rule is of an inferior quality compared to fresh-frozen material. Following reviewer's suggestion, we inserted investigation of the effect

of FFPE per XP group on the main analyses in the paper (Supplementary Fig. 16). For TMB analysis it seems indeed that there might be a tendency for a smaller TMB in FFPE samples, however no significant differences between FFPE and fresh frozen samples were observed in any of the XP groups. The mutational profiles of the samples from FFPE (or Fresh material) do not tend to group together on the cluster dendrograms (Supplementary Fig. 2,3,15b). We added a discussion of these results to the text (L494-503).

R1Q4 The variant calls (in VCF format preferably) should be made available. Also, sequencing reads should be posted to EGA.

Response: We submitted all the newly generated sequencing data from patients and corresponding filtered VCF files with somatic genetic variants to the EGA archive (accession number EGAS00001006732). FASTQ files from cell line experiments are freely available on NCBI (PRJNA940340) and the corresponding VCF files on the Mendeley Data (DOI: 10.17632/jkjkpvgxyd.1; <https://data.mendeley.com/datasets/jkjkpvgxyd>). This information was added to the Data Availability section (L549-554).

R1Q5 The color palette and symbology for scatter plots are hard to see and understand clearly. Some colors are too similar to tell apart (MEL vs. SCC, for example). I'm very confused with the information coding of Figure 4a and 4b: some cancers are doubly encoded by shapes and colors, but others are not?? The text says "well delineated clusters" but I'm confused about what I'm supposed to be looking at and seeing. Also, "XP-C formed a separate cluster" but I think there are just free XP-C samples? Is that right? They look like they are at one end of a range of points. Calling that a separate cluster seems rather arbitrary.

Response: We thank to the reviewer for this comment and agree that the legend and explanation of the Fig. 4b was not very clear. The figures 4a and 4b present sporadic and XP tumors of different tumor types (cSCC, melanoma, BCC). In the revised version we used shapes to distinguish between different tumors types (SCC, melanoma, BCC) and color codes to distinguish between XP groups and sporadic cancers. We believe that MDS plots better capture the variation in multidimension space than hierarchical clustering but agree that the cluster delineation can be arbitrary. We added ellipses to highlight areas on plot exclusively occupied by the given XP groups (Fig 4a,b). Additionally, we reconstructed hierarchical cladograms to highlight the clusters of XP-C, XP-E and sporadic tumors (Supplementary Fig. 8). In line with this we provided a more accurate description of the MDS plot and added the reference to the cladogram (L225-226, L232-239).

R1Q6 Figure 1f: The authors report reconstruction errors, implying that there are more DNA damage processes happening than just SBS7 from UV. But it's just left dangling there. The reconstructed profiles should be reported, those could give clues to what other signatures might be present.

Response: We added the reconstructed mutational profiles (Supplementary Figure 4). They efficiently reconstituted the sporadic mutational profiles while were not able to reproduce specific features of mutational profiles in XP groups. For example, specific for XP-E equal numbers of mutations from TCA and TCC context, enrichment of mutations from TCT context

in XP-C and relatively flat shape of C>T mutational profile for XP-V. XP-V-specific C>A mutations were not reproduced at all, but false enrichment of mutations from T was introduced to XP-C, XP-A and XP-C groups.

R1Q7 Figure 3c: Are there supposed to be error bars? Are the error bars all so tiny that they are covered by the data points? I wonder if the claim that TC-NER operates out to 50 kb past the TES is a bit overstated because it looks like the signal is pretty much down to baseline by about 30 kb?

Response: Fig. 3c is based on a single experiment with XR-seq in XP-C cell line from (Hu et al. 2015 Genes and Development). To further support our findings, we retrieved and reanalyzed another published XR-seq experiments in XP-C cell line from Chiou et al. 2018 j Biol Chem (n=15) and provided them as a Supplementary Fig. 7. It is problematic to precisely estimate the average distance at which extended TC-NER operates because it depends on length of particular genes, level of their expression and probably chromatin state around the genes. Here we provide conservative and average measures. To be more precise we calculated significance of transcriptional bias 100kb around TESs (Supplementary Fig. 6b). The analysis indicated that for XP-C samples with fully abrogated GG-NER and for sporadic melanoma tumors with the large sample size (n=113) the statistically significant effect can be seen up to 40kb on average across the genes. We modified the Results and Discussion sections accordingly (L194, L198-199, L438-442).

R1Q8 Figure 5b: Colors are flipped?

Response: We modified the legend of figures 5a,b and 4d to make it more clear for the reader.

R1Q9 Figure 5f: Would including SBS16 and 36 in reconstruction reduce the error from Figure 1f to levels comparable to other cancers?

Response: Addition of SBS16 and SBS36 to SBS7a/b/c/d reduces the reconstruction error only for XP-V, but not to the level of sporadic cancers. The reconstructed profiles fail to reconstruct the original XP-V mutational profile (see the figure below).

R1Q10 Figure 5e, f, g: Was there just one clone sequenced from each condition? I would be very cautious about over-generalizing from $n = 1$. Summary information on these sequencing samples wasn't mentioned? Should also do reconstruction with COSMIC signatures.

Response: We greatly appreciated these critics of the reviewer which prompted us to perform additional sequencing. In order to be able to assess statistical differences between the genotypes we sequenced two additional clones from *POLH* wt and *POLH*-KO cell lines treated with UVA and UVC (total $n=3$ per genotype per treatment). We observed strongly significant statistical differences between *POLH* wt and *POLH*-KO cell lines for both UV-A and UV-C treatments. To our knowledge this is the first WGS-based evidence for the UV-induced mutator phenotype of polymerase η deficiency (UVA – 3.9-fold increase; UV-C – 10.5-fold increase) (L293-299; Figures 5e,f,g). The summary information from the cell line experiments was added to the Supplementary Table 2.

We performed reconstruction of the mutational profiles of the RPE-1 cell lines from all the experiments using the COSMIC mutational signatures. This analysis revealed a pattern similar to that of XP skin cancers: namely, for treatment with UV-A and UV-C the reconstruction errors of *POLH*-KO cells were significantly higher than for *POLH*-wt cells. This data was incorporated into the Results (L309-313) and Supplementary Figures (Supplementary Fig. 13).

R1Q11 Figure 7: There should be further analysis showing breakdown of missense vs.

nonsense vs. splice site variants. And also conservative vs. non-conservative substitutions for missense. To get a sense of what fraction of variants are likely to have high impact on protein function.

Response: We performed the suggested analysis and incorporated it into the text of the Manuscript (L363-366) and into the Supplementary Figure 14.

R1Q12 The literature on molecular mechanisms for mutagenic bypass of UV lesions should be discussed and cited more specifically, e.g., roles of pol iota and kappa at CPD, and differing mutagenicity of various 6-4 photoproducts.

Response: The possible backup roles of Pol iota and kappa, in the absence of Pol eta, as well as the different strategies cells use to bypass 6-4PP lesions by other TLS polymerases, are now included in the Discussion (L397-404).

Reviewer #2 (Remarks to the Author): Clinical expert in Xeroderma Pigmentosum genetics

In the study “ANALYSIS OF SKIN CANCERS FROM XERODERMA PIGMENTOSUM PATIENTS REVEALS HETEROGENEOUS UV-INDUCED MUTATIONAL PROFILES SHAPED BY DNA REPAIR” the authors assessed the genomes of 38 skin tumors of 24 xeroderma pigmentosum patients and compared them to sporadic skin cancer genomes (152 tumors). The patient cohort included the four most frequent XP complementation groups (XP-A, XP-C, XP-V, XP-D) as well as XP-E. No tumors of the very rare complementation groups XP-B, XP-F, and XP-G were included. The analysis of the tumor genomes from included patients with deficient NER (XP-A, XP-C, XP-D, XP-E) or translesion DNA synthesis (XP-V) revealed how the absence of different NER components impacts mutation burden, mutation profiles, and topography of mutagenesis after UV irradiation. For example a significant increase in mutations was found in cancers from patients with defective GG-NER (XP-C, XP-E) or deficient polymerase eta (XP-V). CC>TT double base substitutions were found to be particularly enriched in XP-C but depleted in XP-E. In XP-V cancer genomes the authors observed that the 3' nucleotide of a pyrimidine dimer is much more frequently mutated than the 5' nucleotide. The authors propose to name this differential mutagenicity of 3' and 5' nucleotides in intrastrand crosslinked DNA dimers “dimer translesion bias”. Furthermore, the authors describe a phenomenon which they propose to call “extended TC-NER”. This describes a decreased mutation load in early-replicating genomic regions which may at least partially be explained by the activity of TC-NER up to 50 Kb beyond the annotated transcriptional end sites. A relatively high number of mutations in a TA/G context were observed in XP-V tumors and confirmed in vitro after UV-A and UV-C exposure of RPE-1 POLH-KO cells.

Overall the manuscript addresses a relevant question, is clearly structured, and very well written. Besides analyzing patients samples some results were also confirmed by in vitro studies using a RPE-1 POLH-KO and a respective wild type cell line. The comparison of different XP cancer genomes provides novel insights on how the absence of different NER components modulates the genome with particular consequences for the patients. The number of assessed tumors is decent considering the rarity of the disease and conclusions drawn from the analyses are supported by the data. Methods are described in sufficient details.

Considering the complexity of different samples, analyses, and results I strongly encourage the authors to provide an overview figure (or a table) that graphically summarizes the (main) results (observed differences in mutation burden, mutation profiles and so on). Such an overview will help the reader to grasp the gist of the manuscript more quickly and provide a reference to quickly retrieve the most important information.

Response: We thank the reviewer for this useful suggestion and added a scheme as a Table 2.

Specific considerations:

R2Q1 Figure 1a: Are in the left panel only C>T SBS represented or all SBS? Please clarify in the figure legend. What kind of post-hoc test was used to generate the indicated p-values?

Response: We modified the legend of Figure 1a and added the information (L1012-1014).

R2Q2 SCC, BCC and MEL represent sporadic cancers and XP-E/C/A/D/V represent results from all tumors of patients in these groups?

Response: We added clarification to the figure legend (L1012-1014).

R2Q3 Line 116: For consistency authors should call DBS “double base substitutions” instead of “tandem base substitutions” throughout the manuscript.

Response: Corrected.

R2Q4 Line 126: A very similar mutation signature as described here for XP-V and shown in figure 1b has been previously reported for SCC in immunosuppressed patients (Signature 32, DOI: 10.1038/s41467-018-06027-1). This should be mentioned and discussed.

Response: We investigated if SBS32 might be related to our XP-V mutational profiles. C>A mutations in XP-V samples are abundant, context-specific and demonstrate strong transcriptional bias indicating that they originate from purine base lesions, while in SBS32 C>A mutations are minor and do not show systematic transcriptional bias (see figure below). C>T mutations in XP-V originate from pyrimidines while in SBS32 they are probably originate from guanines (which is underlaid by their transcriptional bias in SBS32). T>A and T>C mutations in XP-V samples originate in TTN context but in SBS32 they are mostly coming from GTN contexts.

R2Q5 Figure 7d: The figure legend within the figure shows “C>T Transcribed” for both red and orange bars. Based on the text I assume the label for the orange bar is supposed to be “CC>TT Transcribed”. Please correct.

Response: Corrected.

Reviewer #3 (Remarks to the Author): Expert in computational cancer genomics and mutational signatures

In this study, Yurchenko, Rajabi et al have sequenced a number of skin tumors from patients with XP syndromes (with different causal genes) and analyzed mutation patterns therein to provide evidence for various NER/TLS mechanisms gleaned from mutational variation between the genotypes and across the genome. This is a solid, comprehensive genomics study with several findings (some of which are bit speculative perhaps, in terms of mechanisms) of reasonably broad interest to cancer evolution, genome instability, and DNA repair fields. I have some queries for the authors:

Response: We thank the reviewer for the multiple useful suggestions which helped to significantly improve our manuscript.

R3Q1 WGS coverage is moderate, at 40x, and the tumor purity is not great (estimated at 41% overall). Do they see that coverage is very limiting for mutation calling, given the low purity? I.e. do the samples and/or genome regions therein with lower coverage also have lower mutation burdens? Please comment on possible implications of this to conclusions of the study.

Response: This question overlaps with the R1Q2. Please see our response above.

R3Q2 Gene specific C>T mutation signatures they report „some XP groups demonstrated marked differences for C>T mutations in specific contexts, such as enrichment at TCA in XP-E, TCW in XP-C, or NCY in XP-D“. Are these trends consistent across the (few) tumor samples for each group? Since the groups are small there is a concern that an outlier might create a different mutational signature. It is not immediately clear that Fig 1b or the spectra addresses this (it might be read from Supp Fig 1 with some effort).

Response: We thank the reviewer for this point. We performed hierarchical clustering analysis to better assess the uniqueness of XP group mutational profiles and added the plots to the Supplementary figures and to the text (L129, Supplementary Fig. 2, 3, 8a).

R3Q3 Lines 132-138: does this analysis, comparing with known COSMIC spectra, not make more sense by reporting results separately for SBS7a versus SBS7b vs 7c vs 7d, rather than as I understood with the mean of the four spectra? It was unclear to me what the interpretation of the current result would be.

Response: The majority of mutations in sporadic human skin cancers can be explained by the combinatorial activity of mutational signatures SBS7a/b/c/d but not by any single signature (Alexandrov et al. 2020 Nature). For example, in melanomas the majority of mutations are explained by the mixture of SBS7a and SBS7b with some addition of SBS7c and SBS7d. On the Fig. 1f we can see that the lowest reconstruction error was observed for sporadic melanoma samples (MEL). This can be explained by the fact that for the inference of mutational signatures the majority of skin cancer genomes were represented by melanomas (Alexandrov et al. 2020 Nature). The main idea of our analysis was to estimate whether and how well the known UV-induced mutational signatures can explain the observed mutational profiles in XP skin cancers. The results show that they cannot be reconstructed

with the same accuracy as for sporadic cancers. We clarified this question in the Results section (L139-141).

R3Q4 The RT association analysis considered together with the TAD analysis. These two factors (late/early RT and inactive/active TADs) are highly confounded and thus the two analyses are likely redundant. If they wish to show that TADs are associated with mutation rates independantly of the RT (which I doubt, but am ready to be convinced otherwise) they should condition on one variable while testing association with the other. Ditto for chromatin states.

Response: Association of active and inactive TADs compartments with genomic mutagenesis was recently published by Akdemir et al. (Nature Genetics 2020; PMID: 33020667). The authors tried to show that although the TADs and replication timing are confounded by each other the TADs borders “constitute a better proxy to track mutational load change than replication timing measurements.” We did not intend to additionally test for independence of TADs from Replication Timing (RT). Instead we used RT, TAD borders and chromatin states as definitely related, but nonetheless biologically different categories of genomic regions to test the dependency of mutation rates from epigenetic features across XP cancers. Our analysis revealed consistent patterns across three different groups of genomic regions. We clarified our findings and limitations in the Discussion section (L428-435).

R3Q5 They observe the difference in effect (loss of RT association in mutation rates) between GG-deficient XP-C (partial effect) and GG+TC deficient XP-A/XP-D (full effect). This is somewhat hard to interpret. Would this difference in RT-association between the GG-deficient and GG+TC deficient still be observed if looking only at the nongenic regions?

Response: We thank the reviewer for this comment. Indeed we demonstrated this effect on Fig. 2d but did not explain it enough in the Results section. We observed that in XP-C cancers (GG-NER deficient) the mutagenesis in intergenic and untranscribed strand regions was weakly associated with the replication timing (Fig.2d), but the transcribed strand of the genes was strongly associated. In contrast, GG+TC deficient groups did not show any relation of mutation rates with RT on intergenic or genic regions. Now we extended the Results section (L174-176).

R3Q6 Lines 186-188. „Since early RT regions are particularly gene-rich²¹, we hypothesized that in GG-NER deficient XP groups, decreased mutation load in early RT intergenic regions might be associated with the TC-NER activity beyond gene boundaries“ This is fine, and the analysis that follows is interesting however I worry about mis-interpreting of results by readers if not given enough context. I suspect that in early-replicating regions, there may be depletion of mutations also in non-transcribed regions (i.e. those far from gene boundaries). This should be commented on, or better yet shown with data.

Response: It is was known before and also was shown in our manuscript that early-replicating non-transcribed regions even located far from gene boundaries are depleted in the mutations in sporadic skin cancers. This effect is mostly explained by the activity of GG-NER in open chromatin regions (Fig. 2d). The effect is fully abrogated in GG-NER + TC-

NER-deficient XP-A and XP-D samples. With XP-C samples, where TC-NER is active but GG-NER is fully absent we have a exclusive opportunity to estimate how activity of TC-NER alone influences the mutagenic load in non-transcribed regions. We clarified this sentence (L192) and the Discussion part (L439).

R3Q7 „significant TRB up to 50kb downstream of the furthest annotated transcriptional end sites (TES)“ Please do not provide „up to“ numbers in the text, but rather a median or mean, and some description of the distribution of the TRB length after the TES.

Response: Please see the answer to Reviewer 1 (R1Q7).

R3Q8 Considering the XRseq analysis, and how the XRSeq signal extends past the TES of genes and how this mirrors the TRBs (providing additional evidence for the interesting finding that there is post-TES transcription with consequences on mutation rates). Are the lengths of segments of the 2 measurements (TRB, XRseq) correlated across genes? Can they be used to estimate the length of this post-TES protected segment for each gene?

Response: We thanks the reviewer for this suggestion and believe that such kind of measurements can be performed in lab conditions using XR-seq with deep sequencing and different cell lines after UV irradiation. In our work we have enough data to demonstrate the existence of the extended TC-NER and to estimate the average effect size of this process. Further work can be performed to better understand the details of this phenomena in different biological contexts and under mutagenic stress using for example nascent RNA-seq after UV or other bulky DNA lesions-inducing mutagenic stress. We added this perspective to the Discussion section (L446-452).

R3Q9 The XPE (DDB2) analysis is interesting, using mutational data to support the idea that DDB2 is optional in GG-NER, and estimating the drop in efficiency when there is not any DDB2. However I didn't quite follow what is the value in the analysis „To provide a more detailed view of the mutation difference between XP-E, XP-C, and sporadic tumors, we compared the association of mutation load in each group with the core epigenetic marks...“ that there is difference between XPC and XPE mutants was already established by the 1Mb intervals and the dinucleotide signature, so how does this help further. Are the specific marks mentioned in the text somehow interesting or informative?

Response: Following suggestion of the reviewer we put this figure into supplementary materials (now Supplementary Fig. 9).

R3Q10 The section about the „mystery lesion“ that is likely to be copied over error-free by pol eta: this is quite interesting and although speculative, the notion of the thymine-purine dimers is worth mentioning for sure. The one concern is using the correlations between different parts of spectra as evidence „we measured a Pearson correlation of TG > TT or TA > TT mutations with typical UV-induced (YC>YT or CY>TY) mutations and observed strong correlations in both cases, R=0.78 (P = 0.001) and R=0.99“ – isn't this correlation going to be caused, in part, simply by the different „age“ of each tumor? By default any part of spectrum will correlate to some extent with any other part, and this analysis would need to have controls in place to account for that.

Response: We observed variable fractions of TG > TT mutations per sample (15-34%, average = 27%) and hypothesized that if these mutations are not UV-induced and are product of normal physiological cell process such as purine oxidation the rate of their accumulation should be poorly correlated with the UV-induced mutations which typically occur as multiple bursts and associated with the amount of sun exposure. We found strong correlation between UV-induced C>T mutations and TG > TT which probably can be explained by dependency of both mutation types on UV but, other explanations are also possible. We agree that interpretations of this analysis might be different and discussed it in the text (L282-286).

R3Q11 This whole section is a bit dense; I imagine it will be clear to specialists but perhaps less so for the general reader. Consider supplying extra explanations e.g. „Interestingly, UV-A treated POLH-KO cells harbored 52% of T>A and T>C mutations, while in case of treatment with UV-C, it was only 22% (Fig. 5g).“ explain why is this observation interesting. (also other examples above)

Response: we thank for this notion, moved this part to the Discussion section and extended explanation (L481-487).

R3Q12 The tetranucleotide signature analysis of pol eta (XPV) cancers is nice. However, making the link between the presented asymmetry within the CC and the inserter/extender mechanism may be premature. Could not there be other (more parsimonious) mechanisms that explain this pattern?

Response: The one of the most parsimonious hypotheses which we considered was the deamination of 3' cytosine inside the pyrimidine dimer. It is known that the rate of 3' cytosine deamination inside CPD is 3 fold higher than for 5' cytosine (PMID: 34330711). After analysis we rejected this hypothesis because of: (i) relative rate of 3' to 5' cytosine deamination events and subsequent mutations should be constant between sporadic and XP-V tumors but we observe strong increase only in XP-V (ii) the average ratio between mutation rates from 3' and 5' cytosines in XP-V samples was 46-fold, which exceeds all the estimations of differential cytosine deamination between 3' and 5' cytosines; and finally (iii) the effect of almost the same size was observed for T>A and T>C mutations from TT context (Fig. 4c), which is not subjected to deamination. We added these considerations to the Discussion section (L413-423).

R3Q13 The „protein damaging-effects of mutagenesis“ section feels a bit out of place, and also perhaps a bit obvious in parts -- of course C>A transversions are more impactful than C>T transitions. The damaging effects of CC>TT might be (regardless of ratio to silent) not too important in absolute terms because of rarity of dinucleotide substitutions. In this reviewer's opinion, the one interesting part in this section is how strand bias compares to impact. Trimming the rest would help. Also in addition to ref 27 other prior work should be cited here that looked at impact of various mutation processes due to DNA repair activity.

Response: This comment overlaps with R1Q11 of the Reviewer 1. We added additional references and in request of Reviewer 1 modified this section (L356-366). We agree that CC>TT DBSs are usually rare, but due to their enrichment in early-replicating regions and specific enrichment in XP-C (20% in average) and XP-D tumors (17% in average) as well as

high protein-damaging effect they significantly contribute to the protein-damaging mutagenesis in these groups.

R3Q14 Data availability: the somatic mutation calls (not only the raw sequencing data) needs to be made available to the community upon article acceptance.

Response: We submitted all the newly generated sequencing data from patients and corresponding filtered VCF files with genetic variants to the EGA archive (accession number EGAS00001006732). FASTQ files from cell line experiments are freely available on NCBI (PRJNA940340) and the corresponding VCF files on the Mendeley Data (DOI:10.17632/jkjkpvgyd.1; <https://data.mendeley.com/datasets/jkjkpvgyd>). This information was added to the Data Availability section (L549-554).

R3Q15 Fonts in figures are too small. Also the use of the „eta“ symbol for shared first authorship is a nice touch but will probably need to be changed.

Response: We increased the font of the figures and changed the eta symbol in the authorship assignments.

~end~

REVIEWERS' COMMENTS

Reviewer #1 (Remarks to the Author):

The authors addressed each of the reviewers' critiques well. The new data, analyses, discussion text added in response to the reviews have strengthened the paper significantly. Ideally, there would be more tumors sequenced from more patients, but enrolling patients for these kinds of studies is admittedly challenging. It's a caveat that readers will be aware of, but it shouldn't hold up publication. Congratulations on your good work!

Reviewer #2 (Remarks to the Author):

All my comments have been addressed and changes have been implemented accordingly. I have no further comments or objections.

Reviewer #3 (Remarks to the Author):

The revised manuscript was improved, by clarifications to the text, additional supporting analyses and by introduction of more genome sequencing data from a model cell line. My queries were answered satisfactorily.

I think the the volume of new genomic data generated (particularly considering this is a rare syndrome) and the diverse mechanistic conclusions drawn from the data (even if some are speculative), imply that this study makes a substantial contribution to the field. The rigor of the analyses and the clarity of the presentation are very good.

Reviewer #1 (Remarks to the Author):

The authors addressed each of the reviewers' critiques well. The new data, analyses, discussion text added in response to the reviews have strengthened the paper significantly. Ideally, there would be more tumors sequenced from more patients, but enrolling patients for these kinds of studies is admittedly challenging. It's a caveat that readers will be aware of, but it shouldn't hold up publication. Congratulations on your good work!

Response: We thank the reviewer for the through evaluation of our manuscript. Their comments allowed us to particularly strengthen the work. At the same time we assure the reviewer that we continue the sample collection of skin cancers from XP patients of different groups in order to overcome issues associated with the small number of patients, non-optimal storage conditions, different skin cancer tumor types etc.

Reviewer #2 (Remarks to the Author):

All my comments have been addressed and changes have been implemented accordingly. I have no further comments or objections.

Response: We thank the reviewer for their comments and suggestions on the clinical aspects of XP.

Reviewer #3 (Remarks to the Author):

The revised manuscript was improved, by clarifications to the text, additional supporting analyzes and by introduction of more genome sequencing data from a model cell line. My queries were answered satisfactorily.

I think the volume of new genomic data generated (particularly considering this is a rare syndrome) and the diverse mechanistic conclusions drawn from the data (even if some are speculative), imply that this study makes a substantial contribution to the field. The rigor of the analyzes and the clarity of the presentation are very good.

Response: We thank the reviewer for the appreciation of the work done in responding to their valuable comments and suggestions. We also believe that additional explanations and discussion of our results suggested by the reviewer made the paper easier to read and more comprehensive.